# Multi-scale mapping along the auditory hierarchy using high-resolution functional UltraSound in the awake ferret

Célian Bimbard[1†], Charlie Demene[2†], Constantin Girard[1], Susanne Radtke-Schuller[1], Shihab Shamma[1,3], Mickael Tanter[2*], Yves Boubenec[1*]

[1]Audition Team, Laboratoire des Systèmes Perceptifs CNRS UMR 8248, École Normale Supérieure, PSL Research University, Paris, France; [2]Institut Langevin, ESPCI ParisTech, INSERM U979, CNRS UMR 7587, PSL Research University, Paris, France; [3]Institute for Systems Research, Department of Electrical and Computer Engineering, University of Maryland College Park, Maryland, United States

**Abstract** A major challenge in neuroscience is to longitudinally monitor whole brain activity across multiple spatial scales in the same animal. Functional UltraSound (fUS) is an emerging technology that offers images of cerebral blood volume over large brain portions. Here we show for the first time its capability to resolve the functional organization of sensory systems at multiple scales in awake animals, both *within* small structures by precisely mapping and differentiating sensory responses, and *between* structures by elucidating the connectivity scheme of top-down projections. We demonstrate that fUS provides stable (over days), yet rapid, highly-resolved 3D tonotopic maps in the auditory pathway of awake ferrets, thus revealing its unprecedented functional resolution (100/300μm). This was performed in four different brain regions, including very small (1–2 mm³ size), deeply situated subcortical (8 mm deep) and previously undescribed structures in the ferret. Furthermore, we used fUS to map long-distance projections from frontal cortex, a key source of sensory response modulation, to auditory cortex.
DOI: https://doi.org/10.7554/eLife.35028.001

*For correspondence:
mickael.tanter@espci.fr (MT);
boubenec@ens.fr (YB)

[†]These authors contributed equally to this work

Competing interests: The authors declare that no competing interests exist.

## Introduction

Functional ultrasound imaging (fUS) based on Ultrafast Doppler (UfD) was first introduced in neuro-imaging in 2011 (*Macé et al., 2011*). Using ultrasonic plane wave emissions, this system exhibits a 50-fold enhanced sensitivity to blood volume changes compared to conventional ultrasound Doppler techniques (*Mace et al., 2013*), with a very high acquisition rate (ms) enabling unambiguous discrimination between blood flow and motion artifacts (breathing motion, tissue pulsatility,...) (*Demené et al., 2015*). Relative to fMRI, it also presents substantially higher spatial resolution for cerebral blood flow imaging at the expense of non-invasiveness, greater portability and lower cost, and versatility for awake animal imaging. However, most fUS studies thus far have investigated its sensitivity in capturing coarse-grained sensory responses (*Tiran et al., 2017*; *Osmanski et al., 2014a*; *Gesnik et al., 2017*; *Urban et al., 2014*; *Urban et al., 2015*), or used it to explore indirect in-plane brain connectivity (*Osmanski et al., 2014b*; *Rideau Batista Novais et al., 2016*). Also, while the theoretical spatial resolution of Ultrafast Doppler for high sensitivity mapping of microvascularisation has been shown to be 100 μm for whole brain imaging in rats (*Mace et al., 2013*; *Demené et al., 2016*), the ability of the fUS technique to measure *independent* information on functional brain activity from the cerebral blood volume (CBV) variation maps at such a small scale, that is the truly informative fUS imaging resolution, has remained to date unproven. Here, we demonstrate fUS imaging capability in capturing a *fine-grained* 3D functional characterization of sensory

systems and *direct, long-distance* connectivity scheme between brain structures. Our first goal was to provide such 3D high-resolution functional mapping in the auditory system. However the limited richness of stimuli previously applied in state-of-the-art fUS imaging together with their long duration (typically 10 to 30 s) constituted an obstacle as they would require several days of acquisitions incompatible with in vivo investigations. Moreover, most studies used physiological stimuli (*Macé et al., 2011*; *Gesnik et al., 2017*; *Urban et al., 2015*) or direct electrical stimulations (*Urban et al., 2014*) specifically designed to activate at most the entire sensory structures. We therefore drastically reduced the durations and repetitions of presented stimuli while increasing their diversity to push the sensitivity limits of fUS imaging. Consequently we show that this technique can rapidly produce highly-resolved 3D in vivo maps of responses reflecting precise tonotopic organizations of the vascular system in the almost complete auditory pathway of awake ferrets. We further demonstrate that fUS imaging can provide voxel to voxel independent information (with a functional resolution of 100 µm for voxel responsiveness, 300 µm for voxel frequency tuning), indicative of its high sensitivity. These measurements are repeated over several days in small (1–2 mm$^3$ size) and deep nuclei (8 mm below the cortical surface), as well as across various fields of the auditory cortex. On a broader scale, we describe how fUS can be used to assess long distance (out-of-plane) connectivity, with a study of top-down projections from frontal cortex to the auditory cortex. Therefore, fUS can provide a multi-scale functional mapping of a sensory system, from the functional properties of highly-resolved single voxels, to inter-area functional connectivity patterns.

## Results and discussion

Physiological experiments were conducted in three awake ferrets (*Mustela putorius furo*, thereafter called V, B and S). After performing craniotomies over the temporal lobe, chronic imaging chambers were installed (both hemispheres in one animal, and right hemispheres in the other two) to access a large portion of both the auditory (middle and posterior ectosylvian gyri - resp. MEG and PEG) and visual cortex (in caudal suprasylvian and lateral gyri) (*Figure 1a*). The 3D scan of the craniotomy via Ultrafast Doppler Tomography (*Demené et al., 2016*) revealed the in-depth vasculature of the Auditory Cortex (AC) surrounded by the supra-sylvian sulcus (*Figure 1a and b*). In addition, we were able to detect and image deep auditory-responsive structures such as the Medial Geniculate Body (MGB), the Inferior Colliculus (IC) and the dorsal nucleus of the Lateral Lemniscus (DNLL), as well as visually-responsive nuclei such as the Lateral Geniculate (LGN) (*Figure 1—figure supplement 1*).

In order to reveal the tonotopic organization of the auditory structures, we recorded in each voxel the evoked hemodynamic responses to pure tones of 5 different frequencies by computing the % CBV, defined as the percentage of variation in CBV. We then computed the resultant 3-dimensional tonotopic map (*Figure 1c–e*, *Figure 1—figure supplement 2*). Within a relatively short time (10 to 15 min per slice), we could accurately reproduce the known tonotopic organization of the primary (A1 and AAF in the middle ectosylvian gyrus) and secondary auditory cortex (PPF and PSF in the posterior ectosylvian gyrus) (*Bizley et al., 2005*; *Mrsic-Flogel et al., 2006*; *Nelken et al., 2008*), with a high- to low-frequency gradient in A1, reversing to a low- to high-frequency gradient in the dorsal PEG (*Figure 1c*). We note that the fUS enabled us to map within the challenging deep folds of the ferret auditory cortex, such as the supra-sylvian sulcus (sss) and pseudo-sylvian sulcus (pss). Recordings could be performed in the same slice across days, with a high repositioning precision (error <1 slice, 200 µm in that case), which was within the range of the out-of-plane point-spread function for fUS (*Figure 1—figure supplement 3*). Interestingly we were able to capture inter-individual variability along the transect going from the pss to the sss, consistent with previous work in the ferret (*Bizley et al., 2005*).

Large-scale, 3D functional maps were also recorded in the deep and smaller structures of the auditory thalamus (MGB, *Figure 1d*), the inferior colliculus (IC, *Figure 1e*) and the DNLL (*Figure 1e*). The 3D views obtained in fUS allowed us to describe for the first time the tonotopical organizations of the ferret ventral division of the MGB and DNLL. This is particularly remarkable in the latter structure in which we characterize a precise tonotopic map despite its small size (~1 mm-long) and subcortical position (8 mm deep below brain surface). Moreover, such a large field of view allows one to measure simultaneously the functional organization of any coplanar structure (such as A1 and the MGB here), thus opening the door to precise, frequency specific (thalamo-cortical) connectivity

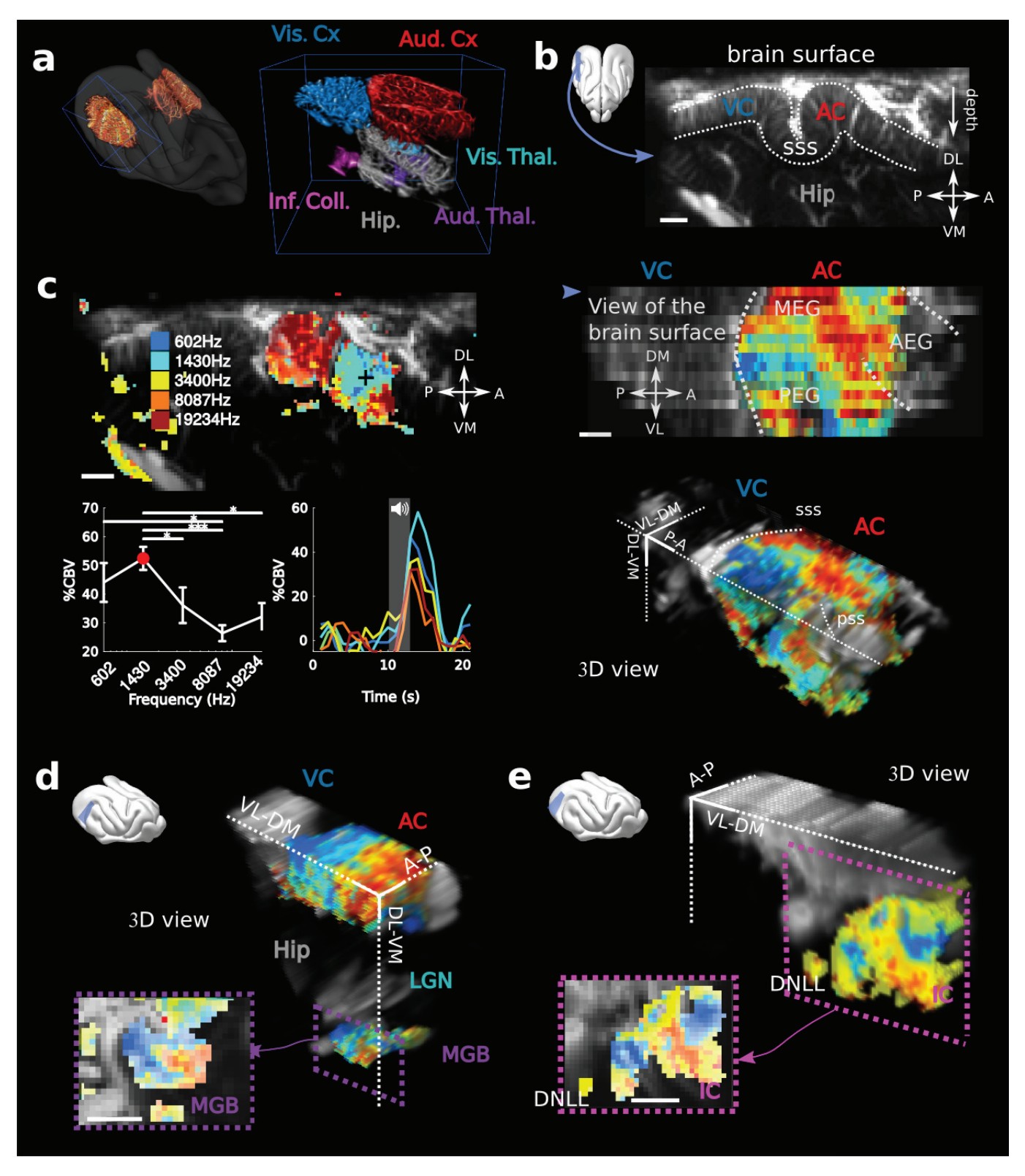

**Figure 1.** fUS imaging reveals the tonotopic organization of cortical, sub-cortical, and intracortical auditory structures in the awake ferret. (a) Left: UFD-T of the left and right craniotomies, superimposed on an MRI scan of a ferret brain. Right: magnification of the blue bounding box (left). Auditory structures: auditory cortices (AC), medial geniculate body (MGB), inferior colliculus (IC). Other structures: hippocampus (Hip), visual cortex (VC). (b) Structural view of a tilted parasagittal slice (~30° from D-V axis) of the visual and auditory cortices (represented as a blue plane on the 3D brain). Lining

*Figure 1 continued on next page*

*Figure 1 continued*

delineates the cortex. (**c**) Upper left: Tonotopic organization of the slice described in (**b**). Lower left: tuning curve (mean ± sem) and average responses in %CBV (see Materials and methods) for the voxel located in the upper panel (black cross). Upper right: combination of 16 similar slices over the surface of the AC, arrow depicts slice of (**b**). AEG/MEG/PEG: anterior/middle/posterior ectosylvian gyrus. Lower right: 3D reconstruction of the whole AC's functional organization. (**d**) 3D reconstruction of both the auditory cortex and auditory thalamus (non-tonotopic areas were masked on this reconstruction for clarity of the representation). Inset: single slice centered on the MGB. Its tonotopic axis runs along the PL-AM axis. Note that (**b–d**) were extracted from the left side of the brain, but flipped for visual clarity and coherence. (**e**) 3D reconstruction of the inferior colliculus and the dorsal nucleus of the lateral lemniscus (DNLL). Inset: single slice centered on the IC. Both (**d**) and (**e**) are tilted coronal slices (~30° from D-V axis). Their tonotopic axis runs along a ~ 20°-tilted D-V axis. All individual and converging scale bars: 1 mm. D: dorsal, V: ventral, M: medial, L: lateral, A: anterior, P: posterior.

DOI: https://doi.org/10.7554/eLife.35028.002

The following figure supplements are available for figure 1:

**Figure supplement 1.** Responses to visual and auditory stimuli in the cortex and thalamus.

DOI: https://doi.org/10.7554/eLife.35028.003

**Figure supplement 2.** Tonotopies in AC, IC and MGB for other animals.

DOI: https://doi.org/10.7554/eLife.35028.004

**Figure supplement 3.** fUS allows for high recording stability and repositioning over days.

DOI: https://doi.org/10.7554/eLife.35028.005

studies. In this respect, future development of high frequency fUS matrix-probes for 3D UfD imaging (*Provost et al., 2015*) will extend this capability to any brain structure.

Single-trial analysis is essential for understanding brain dynamics and behavioral variability. However, it remains a challenge as it necessitates to record high-quality signal from a large number of neurons/voxels at the same time. In order to estimate the reliability and selectivity of fUS single-trial responses, we used MultiVoxel Pattern Analysis (MVPA) to decode the stimulus frequency from the hemodynamic signal. Using a simple linear decoder, we attained high decoding accuracy in the auditory cortex (from 0.46 to 0.63 probability, with chance at 0.2) which was even more striking in the IC and DNLL (from 0.72 to 0.98), despite their smaller size and subcortical location (*Figure 2a*). These results suggest that single trials show reliable and significant activity across all structures.

On a different scale, we sought to demonstrate whether fUS could also reveal encoding differences across cortical layers. We focused on imaging the small vessels in the cortex (keeping only data corresponding to an axial projection of blood flow lower than 3.1 mm/s) and defined cortical layers using an unfolding algorithm providing a flattened version of the AC (*Figure 2—figure supplement 1*). A linear decoder yielded a significantly higher decoding accuracy when using only measurements at intermediate cortical depths (p<1e-3), peaking around 400–500 µm below the surface (up to 0.83, mean 0.67), consistent with it being granular. As a control, we note that baseline blood volume and response magnitude did not show a similar depth-dependent profile (*Figure 2—figure supplement 1*), suggesting that the observed decoding accuracy may be due to variations in capillaries structure within cortical layers (*Adams et al., 2015*). An alternative explanation would be that the improved accuracy at the intermediate depths reflects the underlying neuronal activity, and more specifically the sharper frequency tuning observed in granular layers (*Guo et al., 2012*). Importantly, all these results could be confirmed in single slice recordings, and over several days (*Figure 2—figure supplement 2*), showing that the hemodynamic signal imaged in fUS is reliable enough to decode brain activity on a single-trial basis within a single experiment.

Next, we took a closer look at the tonotopic organization in different structures to examine how tuning curves in neighboring voxels change abruptly. This finding exemplifies the ability of fUS imaging to measure independent information at a very small spatial scale. To quantify the minimal functional spatial resolution of the technique, we defined a discriminability index between voxels, and focused on sharp transition areas (*Figure 2b* left panels). We found that fUS can discriminate responsiveness of neighboring voxels, with a functional resolution as fine as 100 µm (*Figure 2b*). Furthermore, we were able to discriminate voxels based on their tuning curves within a distance of 300 µm in as little as 10 repetitions per frequency (*Figure 2b* and *Figure 2—figure supplement 3*). Importantly, this is a conservative measure of functional resolution, since it largely depends on the smoothness of the underlying functional organization itself (tonotopy) and of the number of trials. The functional resolution described here is thus a lower limit, and could be improved by increasing, for

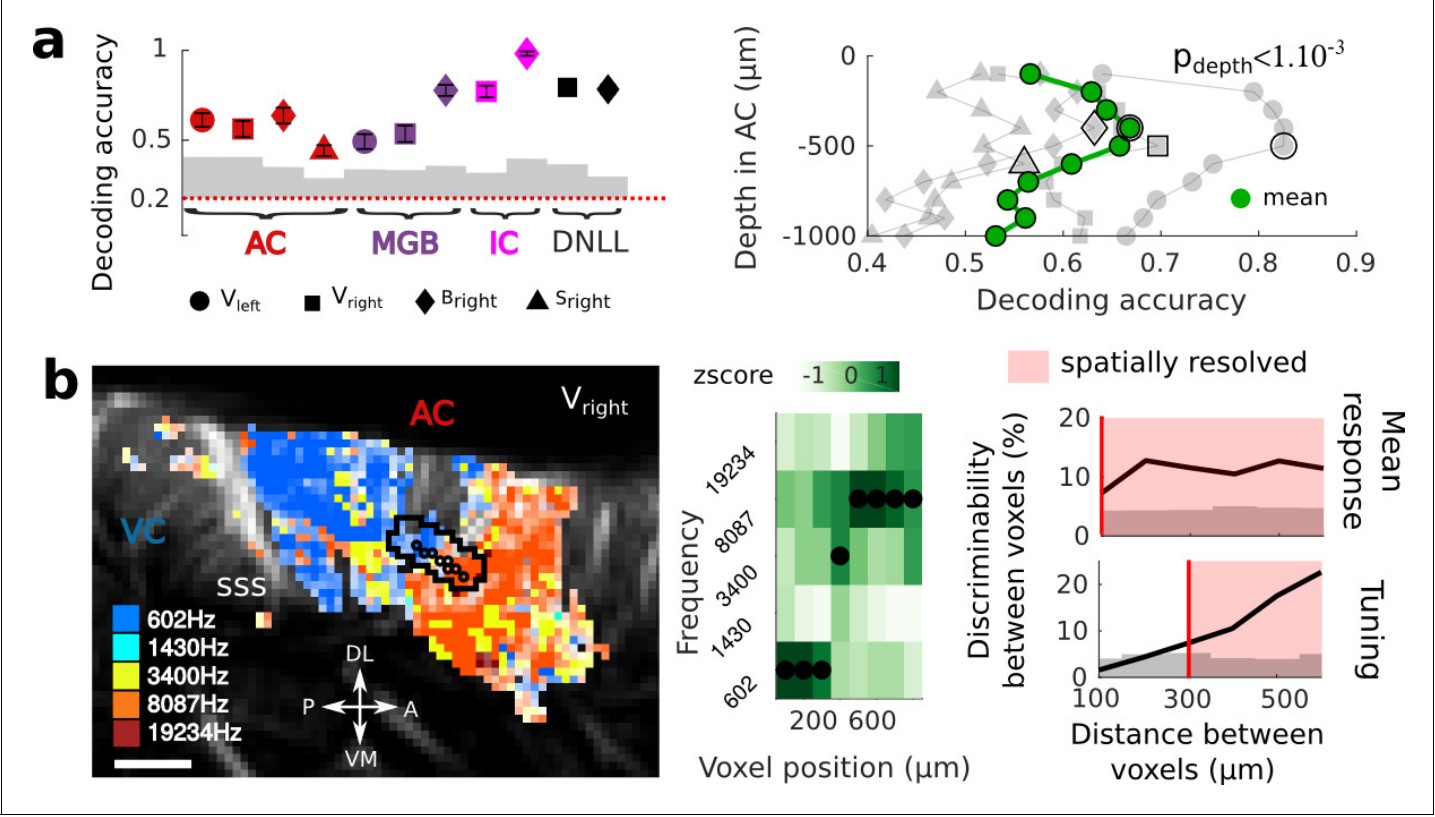

**Figure 2.** Key features of fUS in awake animals: decoding accuracy, layer effect, and effective spatial resolution. (a) Left panel: Decoding accuracy over the five frequencies, in different structures and different craniotomies (see legend). Grey histogram shows the upper limit for chance (p<1e-2, mean ±2 sem computed over 100 randomized decoding sessions). All structures showed significant decoding (p<1e-2). Right panel: decoding accuracy over depths, computed from the activity in the AC of 3 different animals (grey plots). All showed a similar profile, with the accuracy peaking between 400 and 500 μm. The green plot shows the average trend (repeated-measure ANOVA over depth, p<1e-3). (b) Left panel: example of a sharp tonotopic transition from low to high frequency, in the auditory cortex of V$_{right}$ (map not smoothed). Scale bar: 1 mm. Middle panel: heatmap of the z-scored tuning curves of the consecutive voxels (shown by circles in left panel), with the best frequency indicated by a black dot, showing a shift from low to high frequency preference. Right panel: quantification of the lower spatial limit at which one can significantly find differences in the responsiveness (upper) or tuning (lower) of two voxels, with respect to their distance. Grey histogram shows the upper limit for chance (p<5e-2, 5% percentile over 50 randomizations). In that specific case, it was respectively 100 μm and 300 μm. The voxels used in this analysis are the ones within the black contour in left panel, centered on the sharp transition.
DOI: https://doi.org/10.7554/eLife.35028.006

The following figure supplements are available for figure 2:

**Figure supplement 1.** Controls for the decoding across depths.
DOI: https://doi.org/10.7554/eLife.35028.007

**Figure supplement 2.** Single-slice recordings show high decoding possibility on an actual single-trial basis.
DOI: https://doi.org/10.7554/eLife.35028.008

**Figure supplement 3.** Resolution quantification in other regions of the brain, and other animals.
DOI: https://doi.org/10.7554/eLife.35028.009

example, the trial number. These results suggest that fUS can be useful to assess the fine organization of vascular domains within brain structures and to better understand the functional coupling between local neuronal activity and the dynamics of surrounding blood vessels, two important questions for hemodynamic-based techniques (*O'Herron et al., 2016*; *Harrison et al., 2002*).

Another fundamental view of brain function and functional organization is revealed by mapping brain connectivity among various structures. Localizing and quantifying such connections in awake animals, however, remains technically challenging since tracer injections are not an option, and fMRI gives only access to indirect, spatially diffuse measures of connectivity strength. Here, we demonstrate that fUS can be used to probe the functional connectivity between two brain structures that

are far apart: the frontal and the auditory cortices. The frontal cortex (FC) is a region that has been shown to be involved in top-down modulation of early sensory areas, and in particular of the auditory cortex (*Fritz et al., 2003*; *Winkowski et al., 2013*). To reveal its potential links to the auditory areas, we electrically stimulated at different points within the FC while recording evoked hemodynamic responses in the auditory cortex of an awake (slightly sedated) animal (*Figure 3a*). Importantly, this technique does not require any precise priors on the location and nature of the terminal projections. By imaging widely in the auditory cortex, we observed evoked activity in the insular cortex of the pseudosylvian sulcus (PSSC/insula), which was maximal for a certain depth and position of

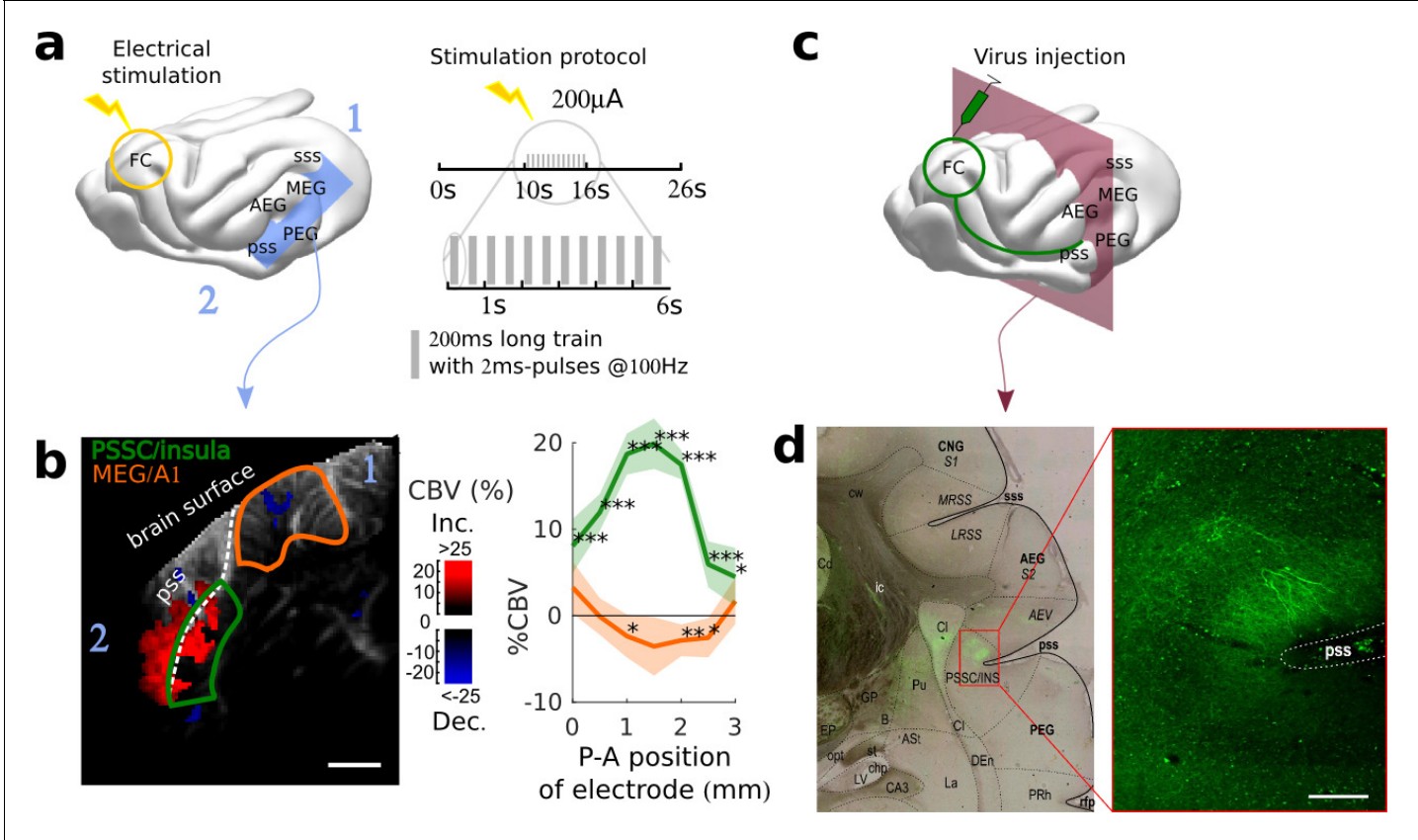

**Figure 3.** Exploring long-distance connectivity: the example of top-down projections from dlFC to the auditory system. (a) Ferret brain with localization of electric stimulation (lightning) and site of fUS imaging shown in (b) (blue plane). A schematic of the electrical stimulation protocol (details in Materials and methods) is also shown in right panel. (b) FC-AC direct projection patterns revealed in fUS. Left: fUS imaging plane along the PSSC/insula, showing modulations of hemodynamic activity in MEG (orange delimitation) and PSSC/Insula (green delimitation) evoked by FC stimulation (map thresholded at +4 sem). The numbers 1 and 2 are here to help orientation. Right: %CBV in the 2 regions of interest after FC electric stimulation (highlighted in the left panel) with respect to the postero-anterior position of the stimulation electrode (0 represents 25.5 mm from caudal crest, 3 represents 28.5 mm), revealing a hot-spot of connectivity at about 1 mm (i.e 26.5 mm from caudal crest) (mean ±2 sem). ***: p-value<1e-3, **: p-value<1e-2, *: p-value<5e-2. Scale bar: 1 mm. (c) Ferret brain with localization of virus (tracer) injection site (green circle) with symbolized projections, and coronal slice represented in (d) (red plane). (d) Anatomical confirmation of connectivity. Left: bright field combined with fluorescence imaging, showing green fluorescent FC projections concentrated in the depth of the PSSC/insula and delineated anatomical structures (scale bar: 200 μm). Right: close-up of the labelled FC projection terminals in the PSSC/insula.
DOI: https://doi.org/10.7554/eLife.35028.010

The following figure supplements are available for figure 3:

**Figure supplement 1.** Frontal Cortex - Auditory cortex connectivity explored further: cortical depth.
DOI: https://doi.org/10.7554/eLife.35028.011

**Figure supplement 2.** Frontal Cortex - Auditory cortex connectivity explored further: secondary areas.
DOI: https://doi.org/10.7554/eLife.35028.012

**Figure supplement 3.** Frontal Cortex - Auditory cortex connectivity explored further: sound and vision.
DOI: https://doi.org/10.7554/eLife.35028.013

the stimulating electrode (*Figure 3b*, *Figure 3—figure supplement 1*). By contrast, there was no evoked activity recorded in secondary auditory areas such as the PEG (*Figure 3—figure supplement 2*). We also observed a decrease in blood volume in the MEG, possibly originating from polysynaptic connections between FC and A1 (*Logothetis et al., 2010*; *Klink et al., 2017*). From these recordings, we cannot disentangle orthodromic versus antidromic activation. We therefore anatomically confirmed the existence of such descending projections from FC to PSSC/insula with independent anterograde virus injections in FC. These injections revealed monosynaptic projections that targeted the PSSC/insula (*Figure 3c–d*), consistent with a contribution of direct projections from FC to A1 to the functional connectivity pattern revealed by the fUS approach. We also observed FC projections in the Claustrum (Cl in *Figure 3c*), ventro-medial with respect to the PSSC/insula. Because the neighboring regions have been reported to be multimodal (*Bizley et al., 2007*; *Bizley and King, 2008*), we subsequently explored the responsiveness of the FC-targeted PSSC/insula to acoustic and visual stimuli. We found this region to be less responsive to broadband noise than A1 (~5% instead of 15%), and not driven by visual stimuli (*Figure 3—figure supplement 3*). Altogether, this experiment offers a proof-of-concept of how fUS can serve as a tool to characterize large-scale functional connectivity without sacrificing any resolution. We can point out two key applications building up on such experiments. First, one may explore connectivity changes in animals, for example during different brain states (e.g., sleep vs. awake), or during the course of learning. Second, and maybe even more importantly, the use of optogenetics can allow a precise mapping between brain structures, targeting for example specific neuronal subpopulations, or projection patterns. The development of such tools has just started, but has been so far limited to fMRI (*Lee et al., 2010*).

To conclude, we have shown that fUS imaging can serve as a technique to record in awake animals a very stable (over days), high-resolution and simultaneous tonotopic mapping of various brain regions, be they large, small, superficial, or deep. This was done over multiple scales, from functional tuning of individual voxels to large-scale connectivity between brain regions. The amplitude of the fUS responses (~20% in the ferret, and close to 50% in neonates [*Demene et al., 2017*]) is quite large compared to typical auditory cortex BOLD responses in fMRI (~5%). This makes mapping both rapid, compared to the electrophysiological approach with multiple penetrations (*Bizley et al., 2005*; *Mrsic-Flogel et al., 2006*), and precise, as illustrated by the ease with which single-trial information can be decoded from its high-sensitivity signal, a key feature when it comes to recording in behaving animals. Furthermore, fUS can be a valuable tool in acquiring broad, yet accurate views of the functional organization of unmapped brain regions and their connectivity with the rest of the brain. Finally, fUS imaging can be readily adapted to mobile and highly stable configurations (*Sieu et al., 2015*), which will make it ideally suited for behavioral cognitive neuroscience studies requiring extended observations, as in the characterization of the neural correlates of learning.

## Materials and methods

### Animal preparation

Experiments were approved by the French Ministry of Agriculture (protocol authorization: 01236.02) and strictly comply with the European directives on the protection of animals used for scientific purposes (2010/63/EU). To secure stability during imaging, a stainless steel headpost was surgically implanted on the skull and stereotaxis locations of the dorsolateral frontal cortex (FC) and the auditory cortex (AC) were marked (*Atiani et al., 2014*). Under anaesthesia (isoflurane 1%), four craniotomies above the auditory cortex were performed on three ferrets ($V_{right}$ and $V_{left}$, $B_{right}$, and $S_{right}$), using a surgical micro drill, yielding a ~ 15×10 mm window over the brain. After clean-up and antibiotic application, the hole was sealed with an ultrasound-transparent TPX cover, embedded in an implant of dental cement (*Sieu et al., 2015*). Animals could then recover for one week, with unrestricted access to food, water and environmental enrichment.

For fUS imaging, animals were habituated to stay in a head-fixed contention tube. The ultrasonic probe was then inserted in the implant and acoustic coupling was assured via degassed ultrasound gel. Experiments were conducted in a double-walled sound attenuation chamber. All sounds were synthesized using a 100 kHz sampling rate, and presented through Sennheiser IE800 earphones (HDVA 600 amplifier) that was equalized to achieve a flat gain. Stimulus presentation were controlled by custom software written in Matlab (MathWorks) and available on a bitbucket repository at this

link: https://bitbucket.org/abcng/baphy/branch/abcng (*Boubenec, 2018*; copy archived at https://github.com/elifesciences-publications/baphy-branch-abcng/).

## Ultrafast doppler imaging

We used a custom miniaturized probe (15 MHz central frequency, 70% bandwidth, 0.110 mm pitch, 128 elements) inserted in a four degree-of-freedom motorized setup. The probe was driven using a custom fully-programmable ultrasonic research platform (PI electronics) and dedicated Matlab software. Ultrasound codes are all are available within the framework of research collaboration agreements between academic institutions.

### 3D vascular imaging

Vascular anatomy of the brain portion accessible from the craniotomy was imaged in 3D using the Ultrafast Doppler Tomography (UFD-T) strategy described in (*Demené et al., 2016*). Briefly, this method acquires 2D Ultrafast Power Doppler (UfD) images at a frame rate of 500 Hz. Each frame is a compound frame built with 11 tilted plane wave emissions (−10° to 10° with 2° steps) fired at a PRF of 5500 Hz, combined with mechanical translation and rotation, and then post-processed via a Wiener deconvolution to correct for the intrinsic out-of-plane loss of resolution, so that we ultimately recover an isotropic 100 µm 3D resolution. In the end, a 3D (14 × 14 × 20 mm) blood volume reconstruction of the vasculature is obtained (voxel size: 50 µm, isotropic resolution 100 µm). This 3D vascular imaging was performed on each craniotomy, and was used as a local reference framework, specific to the craniotomy, where recording planes could be repositioned over days using correlation methods.

### fUS imaging

fUS imaging relies on rapid acquisition (every 1 s) of ultrasensitive 2D Power UfD images of the ferret brain. For each Power image, 300 frames are acquired at a 500 Hz frame rate (covering 600ms, that is one to two ferret cardiac cycles), each frame being a compound frame acquired via 11 tilted plane wave emissions (-10° to 10° with 2° steps) fired at a PRF of 5500 Hz. Image reconstruction is performed using an in-house GPU-parallelized delay-and-sum beamforming. Those 300 frames at 500 Hz are filtered to discard global tissue motion from the signal using a dedicated spatio-temporal clutter filter (*Demené et al., 2015*) based on a singular value decomposition of the spatio-temporal raw data. Although the ultrafast 2ms temporal resolution is available for the CBV image generation, they are in fact averaged into one CBV image every second to capture the dynamics of the cerebral blood physiological response. Nevertheless, it should be noted that this rapid sampling rate is a key asset to unambiguously cancel any respiratory or tissue pulsatility artifacts (*Demené et al., 2015*) in the final averaged images. Blood signal energy (called Power UfD) is then computed for each voxel (100 x 100 x ~400 µm, the latter dimension, called elevation, being slightly dependent of depth) by taking the integral $\frac{1}{T}\int_0^T s(t)^2 dt$ over the 300 time points (*Mace et al., 2013*). This power Doppler is known to be proportional to blood volume (*Rubin et al., 1994*). A certain band of Doppler frequencies can be chosen before computation of the power using a bandpass filter (in our case a fifth order low-pass Butterworth filter), enabling the selection of a particular range of axial blood flow speeds, that is roughly discriminating between capillaries and arterioles (slow blood flow) and big vessels (fast blood flow). In our study, we set the filtering to better focus on small vessels with axial velocity lower than 3.1mm.s$^{-1}$ when indicated in the text. Power UfD signal was normalized towards the baseline to monitor changes in Cerebral Blood Volume (%CBV).

## Protocol for sensory response acquisition

Auditory responses were studied by playing different sounds through animal earphones during recording of the brain activity via fUS imaging. The protocol for sound presentation is as follows: 10 s of silence (baseline), then 3 s of sound followed by 8 s of silence (return to baseline). Trials were following each other with only a little random jitter in time of about 1 to 3 s, and fUS acquisitions were synchronize with the beginning of each trial.

Visual responses were obtained by playing a flickering red-light stimulus instead of sound, with the same durations of different epochs.

## Localization of the auditory structures

In order to find the boundaries of the auditory structures in the imaged portion of the brain, white noise sound was played (70 dB).

## Mapping of the tonotopic organization of the auditory structures

Auditory structures are known to exhibit tonotopic organization based on extensive physiological and structural studies (in the ferret, see [*Bizley et al., 2005*; *Moore et al., 1983*; *Pallas et al., 1990*; *Versnel et al., 2002*; *Nelken et al., 2004*]). To image these tonotopic maps, we played unmodulated pure tones while recording fUS images at five equally spaced frequencies on a logarithmic scale (602 Hz, 1430 Hz, 3400 Hz, 8087 Hz, 19234 Hz, covering the auditory hearing spectrum of the ferret, at 65 dBSPL). The tones were played in random order, 10 trials/frequency (20 in the animal S.). To obtain the whole tonotopic organization in a 3D volume, this process was repeated in different slices in order to build a 3D stack from successive 2D slices (spaced by 300 μm). Each slice was acquired in ~15 min, thus allowing us to map in 3D the whole auditory cortex within a few hours.

We note that these tone stimuli elicited large and reliable responses in the whole auditory tract despite being unmodulated. This suggests that a variety of other auditory stimuli (such as natural sounds) can be used to elicit stronger responses and hence reveal more organizational properties.

## Frontal cortex stimulation

Frontal cortex (FC) electric stimulations were adapted from previously described protocols (*Logothetis et al., 2010*; *Tolias et al., 2005*). Platinium-iridium stimulation electrodes (impedance 200-400kOhms, FHC) were positioned in the region in between the anterior part of the anterior sigmoid gyrus and the posterior part of the proreal gyrus using stereotaxic coordinates, obtained from functional recordings in behaving animals (AP: 25.5–28.5 mm (0 to 3 mm on *Figure 2d*) from caudal crest, caudal crest antero-posterior position being defined at 5 mm lateral from the medial crest/ML: 2 mm (*Radtke-Schuller, 2018*)). Each trial consisted of 10 s of baseline, then 6 s of monophasic stimulation at 100 Hz and 200 μA (2 ms pulses, 200ms-long train, repeated at 2 Hz), after a return to baseline of 10 s. The %CBV was computed as the mean response between 3 and 6 s after stimulation onset. 30 trials were performed for each A-P position of the electrodes. In these connectivity experiments, the animal was slightly sedated using a small dose of medetomidine (Domitor 0.02 mL at 0.08 mg.kg$^{-1}$) to reduce movement artifacts. Stimulation experiments were performed in one ferret, and each of the four experiments presented (*Figure 3* and its figures supplements) was done once, on different days.

## Anatomical tracers

A one year old female ferret weighing 620 g received a 2 μl injection of pAAV2.5-CaMKIIa-hChR2 (H134R)-EYFP (PennCore) as anterograde tracer into left FC. Six months later the animal was perfused and the brain was cryoprotected, shock frozen and cut on a cryostat into 50 μm thick frontal sections into parallel series of which one was counterstained with neutral red. For overview images, combined brightfield and fluorescence images were taken with a Hamamatsu slide scanner 2.0HT (Institut de la Vision) (*Figure 2e*, left). For details, fluorescence images were taken with a virtual slide microscope (VS120 S1, Olympus BX61VST) at 10× magnification (*Figure 2e*, right). Anatomical structures were reconstructed in accord with the ferret brain atlas (*Radtke-Schuller, 2018*).

## Signal processing, analysis and statistics

### Tonotopic maps

Power UfD signal normalized towards the baseline was used to monitor changes in Cerebral Blood Volume (%CBV). The %CBV varied after stimulus presentation (*Figure 1c*) and we quantified voxel responses with the mean of %CBV in a time-window 3 to 5 s after sound onset. Tonotopy of the imaged structures was mapped as follows: for each voxel this mean vascular response across the five tested frequencies was used to determine its best frequency (BF). Statistical differences of the responses to different frequencies in an individual voxel (*Figure 1c*, tuning curve) were assessed using a Wilcoxon rank sum test (post-hoc test after significant ANOVA p<1e-3). For visualization purpose, maps were thresholded by showing only voxels that had (i) a minimal 15% response and (ii) a mean response at their BF highly correlated (p<1e-3) with the mean hemodynamic response. This

thresholding method was used to highlight sound-responsive voxels (disregarding of frequency tuning), and thus allows for the display of zones that were poorly tonotopic (such as AEG). Note here that this thresholding was used only for visualization purposes. Maps constructed with a threshold based on frequency tuning gave similar qualitative results. The mean hemodynamic response was used to approximate the typical vascular response to stimulus (as the Hemodynamic Response Function does for fMRI) and was computed in each structure as the average response over all the voxels showing a response to sound with z-score >3. Note that thresholds could be adjusted depending on the overall responsiveness of different structures and different animals, for illustration purpose. Intriguingly, two additional ferrets did not show any reliable response to sound (responses below 10 %CBV), for unknown reasons. They were not used in the experiments.

Last, maps were spatially smoothed with a $3 \times 3 \times 1$ voxel gaussian filter (std = 0.5), and a 3D median filter ($3 \times 3 \times 3$) was applied to the significance map to remove isolated voxels. The view of the brain surface (*Figure 1c*) was computed as the mean BF averaged from 5 to 10 voxels from the auditory cortex surface delimited manually. For 3D reconstructions of the cortex only, manually adjusted masks were used in order to show only tonotopic regions, and avoid crowdy representations caused by voxel transparency in the 3D visualization. Cortical depths were obtained by manually tracing the surface (just below the pia's blood vessels) and depth limits of the cortex. The 10 different depths were then automatically extracted by a custom-made algorithm (*Figure 2a* and *Figure 2—figure supplements 1* and *2*). The number of voxels at each depth was then equalized for the decoding analysis.

For the single slice analysis presented in *Figure 2—figure supplement 2*, the protocol was designed to speed up tone-responses acquisition (2 s tone, and random interval of 4 to 6 s - uniformly distributed - between two tone presentations). We then used a General Linear Model (GLM) to compute impulse responses of individual voxels to each tone frequency, without any predefined hemodynamic response function. This allowed us to present more stimuli (75 per frequency) in a relatively shorter time (~45 min).

## Decoding

Frequency selectivity of the auditory cortex was assessed using a 5-class linear classifier and a leave-one out strategy: for each frequency pair, vascular responses of the two frequencies (%CBV averaged over 4 to 5 s after sound onset) were separated in a voxel-based space via a linear boundary optimized on 9 of the 10 trials in a learning set. No thresholding procedure was used in this analysis. Overall, pseudo-populations were built by grouping, across all slices recorded within the same structure, trials with identical frequency labels. The decoder was run over 100 shuffles of these pseudo-populations, where train and test sets were randomly chosen. In single slice analysis (*Figure 2—figure supplement 2*), we used a Fisher decoder (normalized by covariance) in order to take into account the noise correlation between voxels in decoding analysis. This was doable thanks to the higher number of tone presentations that allowed us to have a stable estimation of the covariance matrix.

In order to prove the significance of the obtained accuracy, we used a permutation procedure in which we shuffled the labels (i.e., which frequency was played during each trial) across trials, and performed the same decoding analysis, thus obtaining the chance distribution for decoding accuracies. We used 100 permutations, and considered that the real decoding accuracy was significantly out of the chance distribution (trial frequency labels shuffled) when above the 95th percentile. All the actual decoding accuracies were above the chance decoding accuracies. Our p-value resolution is limited by the number of permutations (100) and therefore our obtained p-values are all below 0.01.

To evaluate whether cortical depth had an effect on decoding accuracy (*Figure 2a*), we performed a one-way repeated-measure ANOVA over the four different craniotomies, with depth as the factor.

## Resolution quantification

In order to quantify the minimal spatial scale at which fUS can provide independent information from two neighbouring voxels, we focused on sharp edges of functional transition and performed 2-way (voxel and frequency as factors) ANOVA on the tuning curves (%CBV averaged over 4 to 5 s

after sound onset) of each pair of voxels within a certain contour (example transect and contour shown in *Figure 2b*, left panel). The voxel factor quantified the dissimilarity in the average responses for two voxels, being thus representative of an overall responsiveness dissimilarity when significant. The interaction term (frequency x voxel) quantified how dissimilar the tuning curves were for two different voxels, independently of their overall responsiveness. This term therefore represented our ability to discriminate between different functional voxel tuning. Pairs of voxels were considered to be 'dissimilar' (in responsiveness or tuning) when the associated p-value was $<5.10^{-2}$. Importantly, these values depend on the smoothness of the underlying functional neuronal map (the sharper the better) and on the number of trials used in each experiments (the higher the better). Here, we show that using only 10 trials per frequency, we could go down to a functional resolution comparable to the voxel size (100 µm) for the overall responsiveness, and of 300 µm for the tuning.

We randomized 50 times the responses over all voxels and all frequencies and performed the same analysis to find the average distribution expected by chance for both responsiveness and tuning dissimilarity percentages. We determined the spatial resolution as the shortest distance between two voxels at which the actual number of dissimilar pairs was above the 95th percentile of the randomized distribution. Distance between voxels defined by coordinates $(x_1,y_1)$ and $(x_2,y_2)$ was computed as the rounding of $\sqrt{(x_1 - x_2)^2 + (x_1 - x_2)^2}$.

Finally, we performed this analysis in different regions (AC and IC) and different animals ($B_{right}$, $V_{left}$, $V_{right}$, $S_{right}$) in order to generalize this result (*Figure 2—figure supplement 3*).

## Acknowledgements

We thank Marc Gesnik from the Institut Langevin for his valuable inputs for the ultrasound sequence programming, Roberto Toro for the ferret fMRI scan, and Kishore Kuchibhotla for careful reading of the manuscript. This work was supported by ANR-10-LABX-0087 IEC et ANR-10-IDEX-0001–02 PSL* and research grants from the European Research Council under the European Union's Seventh Framework Program (FP7/2007-2013)/ERC Advanced grant agreement n° 339244-FUSIMAGINE and ERC Advanced grant agreement n° ADG_20110406-ADAM and R01-DC005779 (SS). The project received the technical support of the INSERM Technology Research Accelerator in Biomedical Ultrasound.

## Additional information

### Funding

| Funder | Grant reference number | Author |
| --- | --- | --- |
| European Research Council | ADG_20110406-ADAM | Célian Bimbard<br>Constantin Girard<br>Susanne Radtke-Schuller<br>Shihab Shamma<br>Yves Boubenec |
| Agence Nationale de la Recherche | ANR-10-LABX-0087 IEC | Célian Bimbard<br>Constantin Girard<br>Shihab Shamma<br>Yves Boubenec |
| Agence Nationale de la Recherche | ANR-10-IDEX-0001-02 PSL* | Célian Bimbard<br>Charlie Demene<br>Constantin Girard<br>Susanne Radtke-Schuller<br>Shihab Shamma<br>Mickael Tanter<br>Yves Boubenec |
| European Research Council | 339244-FUSIMAGINE | Charlie Demene<br>Mickael Tanter |
| National Institutes of Health | R01-DC005779 | Shihab Shamma |

The funders had no role in study design, data collection and interpretation, or the decision to submit the work for publication.

## Author contributions
Célian Bimbard, Conceptualization, Data curation, Formal analysis, Validation, Investigation, Visualization, Methodology, Writing—original draft, Writing—review and editing; Charlie Demene, Software, Visualization, Methodology, Writing—review and editing; Constantin Girard, Formal analysis, Investigation; Susanne Radtke-Schuller, Formal analysis, Investigation, Visualization; Shihab Shamma, Funding acquisition, Writing—review and editing; Mickael Tanter, Conceptualization, Resources, Funding acquisition, Methodology, Writing—review and editing; Yves Boubenec, Conceptualization, Resources, Supervision, Funding acquisition, Investigation, Methodology, Writing—original draft, Project administration, Writing—review and editing

## Author ORCIDs
Célian Bimbard (iD) http://orcid.org/0000-0002-6380-5856
Yves Boubenec (iD) http://orcid.org/0000-0002-0106-6947

## Ethics
Animal experimentation: Experiments were approved by the French Ministry of Agriculture (protocol authorization: 01236.02) and strictly comply with the European directives on the protection of animals used for scientific purposes (2010/63/EU). All surgery was performed under anaesthesia (isoflurane 1%), and every effort was made to minimise suffering.

## Decision letter and Author response
Decision letter https://doi.org/10.7554/eLife.35028.020
Author response https://doi.org/10.7554/eLife.35028.021

## Additional files
### Supplementary files
• Transparent reporting form
DOI: https://doi.org/10.7554/eLife.35028.014

### Data availability
The data that support the findings of this study can be found at https://lsp.dec.ens.fr/en/research/supporting-materials-848. The full raw imaging files are >20Tb and are therefore available on request to the corresponding author.

The following datasets were generated:

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
