## [Decision Letter]

Thank you for submitting your work entitled "Multi-scale mapping along the auditory hierarchy using high-resolution functional UltraSound in the awake ferret" for consideration by *eLife*. Your article has been reviewed by four peer reviewers, and the evaluation has been overseen by a Reviewing Editor and a Senior Editor. The following individual involved in review of your submission has agreed to reveal their identity: Victoria Bajo Lorenzana (Reviewer #1).

Our decision has been reached after consultation between the reviewers. Based on these discussions and the individual reviews below, we regret to inform you that your work will not be considered further for publication in *eLife*.

As you will see from the comments included below, the four reviewers differed in their opinions on your paper. Although these differences were not fully resolved in the ensuing discussion, it was agreed that previous studies (e.g. Gesnik et al., 2017) have demonstrated the feasibility of using functional ultrasound imaging to measure responses in cortical and subcortical structures. We recognize that there are clear advances over that study in the present work, including imaging at higher spatial resolution and the use of awake animals. However, other studies have employed functional ultrasound imaging in awake animals, so the question of novelty for a methods paper is something that we have to consider carefully. A Tools and Resources paper not only needs to describe a significant methodological advance, but also to do so in a way that would allow others to adopt the technique in their own work. All the reviewers agreed that your paper does not achieve this and that there are numerous areas where the experimental and/or analytical details provided are inadequate. To a large degree, these points are potentially addressable, but other concerns raised included the variable maps obtained from different animals, some inconsistencies with previous electrophysiological or intrinsic imaging studies of ferret auditory cortex, the time course over which the signals were obtained, and the effects of electrical stimulation of frontal cortex, which were regarded as not particularly convincing. Although comparisons are drawn with published data obtained using other methods, it was felt that independent validation of the frequency tuning using, e.g. microelectrode recording would be desirable.

On the basis of the reviews, we will unfortunately not be able to publish your work as a methods paper in the Tools and Resources section of *eLife*. However, we recognize the value of this interesting approach for measuring activity in different brain regions of the same animal and would therefore welcome future submissions in which you use functional ultrasound imaging to investigate specific questions in the auditory system.

Reviewer #1:

The authors use Functional Ultrasound (fUS) to image the auditory brain of awake ferrets with milliseconds temporal resolution and 100 µm spatial resolution. fUS is based on ultrafast Doppler but using ultrasonic plane wave emissions claiming a 50 fold enhanced sensitivity to blood volume changes.

Results in three animals (one imaging both hemispheres) show tonotopy in the auditory cortex (MEG, PEG), auditory thalamus (MGB), inferior colliculus (IC) and lateral lemniscus (LL). They also show blood volume changes in the auditory cortex following electrical stimulation of the frontal cortex in a single experiment.

Imaging of tonotopic arrangement of different auditory structures seems convincing, whereas the activation of the PSSC/insula by electrical stimulation of the frontal cortex looks vague. Details about how many cases, what place in the frontal cortex and electrical settings were stimulated most effectively must be added. The effect of electric stimulation of frontal cortex in the Claustrum needs to be reported. It has not been reported when the connection is clear (Figure 3C), but not a change in blood volume unless the increase observed in Figure 3B includes both PSSC and Claustrum together. In fact, the increase is observed also dorsal to the PSSC and a mix of increase and decrease deeper than that (Figure 3—figure supplement 1). Whether a change in blood volume after far away electrical stimulation is indicative of connectivity or otherwise need to be clarified. Results and Discussion section. Please state where exactly was the stimulating electrode; to say that the evoked activity in the PSSC/insula was maximal for a certain depth and position of the stimulating electrode is imprecise. The decrease of blood volume in MEG is indicating some kind of polysynaptic inhibition? If PSSC/insula area is not driven by sensory stimulation (broadband noise or visual stimulation), is suggested that is not a sensory structure?

Comparison in tonotopy is tricky across cases. Figure 1 is the best example but high frequencies in MEG are not only in the tip of the gyrus but also in between A1 and AAF. The other two cases were very different with few high frequencies in V_right_ and AAF full of high frequencies in Bright case (Figure 1—figure supplement 2). Also, in these two cases the tonotopy in IC and DNLL look less convincing than in Figure 1. The authors should explain in detail the different cases and discuss the differences between cases.

The paper will improve if further details are added. For example, how many repetitive days of imaging is not clear until Figure 1—figure supplement 3 (even here it is difficult to read the y-axes legend). Where exactly was the electric stimulation in the frontal cortex was applied: anterior sigmoid gyrus, lateral part, at what depth? LL is related to lateral lemniscus fibres, lateral lemniscus nuclei, only the dorsal nucleus of the lateral lemniscus as suggested in one of the figures; the patterns of sensory and electrical stimulation are not clear; 2 s stimulation every (8 + 10 s) of silence or only 8 s of silence? In this 2 seconds of stimulation, how long is the stimulus? The% CBV as percentage of cerebral blood volume is not explained until subsection “Signal processing, analysis & statistics”.

The time resolution of the technique seems to be more in the range of seconds than milliseconds when the quantified voxel responses are taken in a time-window 3-5 seconds after sound onset. Could the authors explain it?

Red and blue color code in figures is extremely confused. It can apply to increase and decrease in blood volume, to auditory cortex and visual cortex, or even high frequency and low frequency of auditory stimulation. I suggest the use of different colors when possible and always add a color code scale to each panel.

The decoder accuracy is very poor, at least in the auditory cortex and MGB (Figure 2). It would be good to add a plausible explanation about the fact that is better in the middle layers of the cortex (about 500 microns).

This technique could be a great complement to another imaging and recording techniques used in parallel with behavior in awake preparation. It would add value to include a new section of future potential applications where limitations were also discussed, for example having the head fixed or the need for sedation.

Reviewer #2:

1) As the key novelty of this paper critically relies on the multivoxel pattern analysis/decoding accuracy and discriminability/resolution index calculations, these analysis procedures need to be described much more carefully and the numerous ad hoc thresholds currently described in the analysis section appropriately justified. How the comparison for multiple corrections is done also needs clarifying. If this work is meant to speak to the neuroimaging community at large, it needs to employ standard neuroimaging analysis as well as provide a link between its output and the metrics currently employed.

2) Given that both spatial coverage and resolution reported critically depend on the extensive skull removal, the limitations of this technique as a means of studying across-region connectivity needs to be carefully qualified. This also makes current comments on how the technique compares to fMRI resolution ill grounded: if invasiveness is allowed, then implantation (even on pial surface!) of RF coils also affords much higher spatial resolution than what the authors currently quote.

3) The rigor of figure creation is subpar: all of the images that have colored overlays need to have a color bar alongside with numbers reflecting the mapping of those colors to actual values of physiologically interpretable quantities. The reported CBV changes are very high compared to the literature, even in the awake mammals, so a discussion on this topic is warranted as well.

Reviewer #3:

The paper titled "Multi-scale mapping along the auditory hierarchy using high-resolution functional UltraSound in the awake ferret" – submitted as a Tools and Resources article in *eLife* – reports imaging of tonotopic maps in the awake ferret and imaging of activity evoked in auditory areas by electrical stimulation of the frontal cortex in the sedated ferret, with functional ultrasound imaging. The authors present these results as a new method for recording brain activity at multiple spatial scales at higher resolution in awake animals, broadly applicable for other neuroscience questions. The imaging of tonotopic maps could be of interest for the auditory field, particularly in deeper structures difficult to access. However, there are major problems concerning both the novelty and the relevance of these experiments to justify the claim made by the authors in term of methodological advances. For this reason, I do not recommend the publication of this article as a Tool and Resources article in *eLife*.

Major comments:

1) There is no significant technological or methodological advance compared to previous work. The authors claim that they did for the first time (a) in an "awake" animals (b) "multi-scale mapping" (c) at "high-resolution". These three points have been shown before with the same technique.

a) There were three papers describing functional ultrasound imaging, including two papers in awake freely-moving rats (Macé et al., 2011; Urban et al., 2015 and Sieu et al., 2015) (one was not cited by the authors).

b) The possibility to record at "multiple spatial scales" (defined by the authors as the possibility to image mesoscale patterns of activity within a brain region and across brain regions) has been shown previously. For example, a single barrel was mapped in the rat brain, as well as large scale activity during epileptic seizures (Macé et al., 2011). Odor topographic maps have been imaged within the rat olfactory bulb (different glomeruli) and piriform cortex (Osmanski et al., 2014). Large-scale connectivity was explored previously in the awake rat (Osmanski et al., 2014). Imaging of sensory responses in small deep subcortical nuclei were shown in freely moving rats (Urban et al., 2015). A recent paper by the authors reported mapping of the visual responses in the rat brain at large-scale as well as local scale (for different retinotopic positions) in visual cortex (Gesnik et al., 2013).

c) The resolution of the technique (100 µm) has been demonstrated experimentally (point spread function) and theoretically in a previous paper (Macé et al., 2017). Therefore, the title "high-resolution functional ultrasound", as well as claims about spatial resolution made in the article are misleading, given that no improvement was made. In fact, the ultrasound acquisition and data processing used in this work is the same as in previous papers (for example in (Gesnik et al., 2013)).

d) Concerning the temporal resolution, the authors claim in the introduction that functional ultrasound imaging has "ms resolution compared to fMRI" (Introduction) and use the term "rapid" multiple times in the manuscript. However, they acquire one image of one brain slice in 1 second (subsection “fUS imaging”), and the whole tonotopic map in several hours (subsection “Protocol for sensory response acquisition”). Although the spatial resolution of fUS is better than fMRI, this is not the case for the temporal resolution.

2) Beyond the problem of novelty, there is a major issue with the claim about the "awake" state in this paper. The second part of the paper, about long-distance connectivity, was done in sedated ferret (subsection “FC stimulation”). This is not clear to the reader except in the Materials and methods section. The paragraph on connectivity experiments starts with: (Results and Discussion section: "Localizing and quantifying such connection in awake animals, remains technically challenging […] Here we show that fUS can be used to probe functional connectivity between two brain structures") leading the reader to think this is done awake. The choice of this experiment, in the context of demonstrating the interest of a new tool for imaging awake ferrets, is disputable. Whereas only tonotopic maps were recorded in awake (restrained) animals, the awake state is emphasized at multiple instances, including in the title and abstract. Moreover, obtaining tonotopic maps do not require awake state (some examples under anesthesia: (Nelken et al., 2008; Bizley et al., 2005 and Mrsic-Flogel, Versnel and King 2006)). A key advantage of functional ultrasound is to be applicable to awake, behaving and freely-moving animals, as shown previously (Urban et al., 2015; Sieu et al., 2015). Although many interesting biological insights can be obtained under sedation, this limits the interest of this work compared what was previously published, in particular for a methodology paper.

3) As an addition to the point 2, the authors report in the Materials and methods section that "the ferret was sedated to avoid movement artefacts". Previous studies were done in freely moving animals without motion problems (Urban et al., 2015; Sieu et al., 2015). Why is motion a problem in connectivity experiments compared to tonotopic experiments is unclear and not discussed. The claim in the conclusion that this method is "readily adapted to mobile and highly stable configurations" is undermined by the fact that the connectivity study was done in sedated animals.

4) Functional ultrasound imaging technique is applied here in a different animal model, the ferret. However little effort is done to provide readers with a protocol for reproducing the work. For example, identification of brain regions is an important step for other users with different questions. Identifying brain regions anatomically on ultrasound images is not easy compared to, for example, MRI. This step is not clearly explained in the article or in the Materials and methods section. It was apparently done manually and/or based on the auditory responses (which is not translatable to other behaviors). The usability for other neuroscientists to study other questions in the ferret, for example to reveal new regions implicated in a given behavior, is therefore limited. Availability of the ultrasound codes for other users is not stated.

5) The connectivity experiment, used to demonstrate top-down projections from the frontal cortex to the auditory system, is conceptually problematic. Electrical stimulation is known to evoke antidromic activity; therefore, it is not possible to discriminate between top-down and feed-forward connections if reciprocal connections exist. Optogenetic activation would have helped resolving this issue (and would have broadened the applicability for other neuroscience questions). Compared to simple tracer injections, it is unclear what we learned from this experiment, in part because the functional data presented are not discussed and all the focus is put on the method.

Comments on Methodology:

- Tonotopic maps (Materials and methods section). The authors display voxels in tonotopic maps if the max response is above a certain threshold (15%) and if the max response is correlated with the average hemodynamic response function (p<1e-3). Then they indicate: "these thresholds are adjusted for different structures and different animals". First, the same thresholding parameters should be used for the three structures of the three animals. Second, the frequency tuning is not taken into account in this thresholding method, only amplitude. Any voxel responsive to sound, regardless of whether it is tuned or not, would pass this threshold. A statistical test should be used on each voxel to determine significant tuning, and only tuned voxels should appear on the maps. Third, the authors applied a mask manually to show only voxels in auditory structures (subsection “Signal processing, analysis & statistics”). This biases the results and is misleading. It is not possible to assess the quality of their thresholding method outside of the auditory regions.

- Significance of the frequency tuning is tested only for one single voxel of one animal (Figure 1C, subsection “Signal processing, analysis & statistics”). This is uninformative. Average tuning with respect to preferred frequency of all voxels would have been the right analysis to do. This should be done for all regions separately, compared to control regions of the same size and compared across animals.

- Displaying an average tonotopy map for each brain region would be beneficial. The organization in each region is not clear from the different examples. In particular, in Figure 1—figure supplement 2, maps of the third animal are different from the other two animals (for example, in PEG). No comparison is made with the known tonotopic maps from the literature (such as: axis of the tonotopic map, reversals/boundaries of the maps) (Nelken et al., 2008; Bizley., 2005 and Mrsic-Flogel, Versnel and King 2006).

- The interest of the multivoxel pattern analysis decoding is disputable. The analysis shows that, when pooling all voxels from one structure, single trial responses carry significant information about the frequency tuning. First, that does not mean that a tonotopic map can be reconstructed at the single-voxel level from a single trial. Second, this kind of analysis is usually used to determine if a property (i.e. sound frequency tuning here) is encoded in a brain region when the spatial resolution is too low or if responses are intermingled. Here it is clear that frequency tuning exists in auditory areas. The claim that " the hemodynamic signal imaged in fUS is reliable enough to decode brain activity on a single-trial basis within a single experiment" (Results and Discussion section) is trivial. It could have been demonstrated in a much simpler way – for example by averaging single-trial tuning curves of all voxels of the tonotopic map in a specific region.

- The quantification of the spatial resolution was made based on 6 voxels hand-picked in one animal and one structure (Figure 2B). It is not reported where (in which brain structure? which animal?) and these voxels are not put in the context of a larger map. This is not sufficient to sustain this claim. Such quantification should be made on more voxels, in different structures and compared across the three animals.

- Statements about the multimodal responsiveness of brain areas (Results and Discussion section, Figure 3—figure supplement 3) are not sufficiently supported. The authors report the data obtained from only one animal. Generalization on the multimodal aspect of these regions would require a statistical test across different animals. Moreover, as mentioned before, the anatomical identification of brain regions should be standardized based on an atlas and not manually picked after the functional recordings. For example: responses in PSSC seem restricted to the upper voxels of the delimited region (Figure 3—figure supplement 3Figure). This could be due to a misalignment of the region boundaries, thus misattributing voxels from the neighboring auditory responsive regions to PSSC.

Reviewer #4:

Overall the study looks impressive and interesting to me, and I have just a few statistical questions.

The primary voxelwise activation response mapping was thresholded both in terms of minimal% signal change and statistically. However, the latter does not appear to have corrected for multiple comparisons across voxels and frequencies. This needs to be done for clarity and transparency. However, I'm not saying that any potential lack of full family-wise-error significance in the initial voxelwise mapping would necessarily cause concern for the later multivariate analyses or overall story.

The exact calculations being made for the resolution quantification need to be written out a little more explicitly and clearly.

For any timeseries analysis involving temporal correlation (whether for external-sensory or electrical stimulation) please describe whether/how temporal autocorrelation was adjusted for in the statistical analyses.

Figure 1A caption – presumably the authors mean structural MRI not fMRI?

[Editors’ note: what now follows is the decision letter after the authors submitted for further consideration.]

Thank you for resubmitting your work entitled "Multi-scale mapping along the auditory hierarchy using high-resolution functional UltraSound in the awake ferret" for further consideration at *eLife*. Your revised article has been evaluated by Andrew King as the Reviewing and Senior Editor and three reviewers.

Although the manuscript is improved, the reviewers had widely differing opinions regarding the extent to which the manuscript has been adequately revised, which we were not able to resolve in the subsequent consultation. In particular, reviewer 3 continued to raise a number of concerns. The key elements of those concerns, along with the minor comments provided by the other reviewers, are summarized in the following. We are hoping that you will be able to address these in a further (and final) revision.

Essential revisions:

The main argument of the authors to justify novelty is that the auditory stimulation (5 tones, 3 s presentation) "pushes the limits of fUS sensitivity". There is no conceptual difference between this paradigm and previous studies that already used fUS to record sensory-evoked activity in rats or other models. The authors argue for shorter stimulation time (3 s), compared to previous studies, as a novel aspect of their paradigm. However, (Urban et al., 2014) already recorded the response to forepaw stimulation as short as 200 µs (see graphical abstract and Figure 4, for example). Yet, this article is not cited. Another claim made by the authors is that they show, for the first time, activations in very small and deep structures. A highly relevant paper (Urban et al., 2015) contains a dedicated section about "Functional imaging in subcortical brain structures", showing, for example, activation in small and deep thalamic nuclei in freely moving rats. However, although this paper was brought to the attention of the authors, they did not add this reference to the manuscript or address these concerns about novelty.

Reviewer 3 stated that you should emphasize that both ultrasound sequence and data processing used in this study are identical to previous studies, and thus, the imaging resolution remains the same. Indeed, a resolution of 100 µm (the physical resolution of the method) was claimed in previous papers, so it remains unclear why this resolution is presented as "unprecedented". Because the same fUS sequence was used here as in previous studies, the apparent improvements in sensitivity and resolution most likely relate to differences in experimental protocol, including the stimulation strategy used. The revised version of the paper addressed some of these points, but not to the satisfaction of reviewer 3. Please take another look at this and ensure that all relevant key studies are cited.

Another concern that has not been well addressed by the authors is their comparison with fMRI. The authors insist on the millisecond acquisition rate of fUS. While the acquisition of a compound ultrasound image (~2 ms) is fast, the relevant information about neural activity is derived from – much slower – images of blood volume (~ 1 s) that are computed from hundreds of ultrasonic images. Thus, the relevant temporal resolution of fUS is on the order of seconds and not few milliseconds. Also, fMRI is not slow compared to fUS; the acquisition of a single slice is usually faster. As an example, (Leaver and Rauschecker, 2016) acquired tonotopic maps using 6 tones of 2 s duration. Each brain slice was acquired in 250 ms and the full volume (28 brain slices) in 7 s. By comparison, a single brain slice is acquired in 1 s with fUS. The advantages of fUS compared to fMRI reside in the higher spatial resolution and ease-of-use, but not in the temporal resolution. Hence, insisting on a superior temporal resolution of the technique is not justified.

Reviewer 3 continues to have a concern about the way the tonotopic maps were constructed, insisting that this should be based the preferred frequency of pixels modulated by the sound frequency. We believe that – for the reasons that you outlined in the response letter – this is not necessary, but please ensure that the manuscript text fully justifies the thresholding procedures used. We do not think it necessary (or desirable) to use the approach illustrated in the response letter, where a threshold of 10% was used. The reviewer proposed that you should average over more trials or use more stimuli to improve sensitivity to modulation. If you have additional data along these lines, that would clearly help to address this concern, but we believe it is useful to show non-tonopically organized regions too and to focus on the MVPA for quantification of the reliability and sensitivity of the responses (albeit with a little more explanation).

[Editors’ note: the author responses to the first round of peer review follow.]

---

## [Author Response]

[…] As you will see from the comments included below, the four reviewers differed in their opinions on your paper. Although these differences were not fully resolved in the ensuing discussion, it was agreed that previous studies (e.g. Gesnik et al., 2017) have demonstrated the feasibility of using functional ultrasound imaging to measure responses in cortical and subcortical structures. We recognize that there are clear advances over that study in the present work, including imaging at higher spatial resolution and the use of awake animals. However, other studies have employed functional ultrasound imaging in awake animals, so the question of novelty for a methods paper is something that we have to consider carefully. A Tools and Resources paper not only needs to describe a significant methodological advance, but also to do so in a way that would allow others to adopt the technique in their own work. All the reviewers agreed that your paper does not achieve this and that there are numerous areas where the experimental and/or analytical details provided are inadequate. To a large degree, these points are potentially addressable, but other concerns raised included the variable maps obtained from different animals, some inconsistencies with previous electrophysiological or intrinsic imaging studies of ferret auditory cortex, the time course over which the signals were obtained, and the effects of electrical stimulation of frontal cortex, which were regarded as not particularly convincing. Although comparisons are drawn with published data obtained using other methods, it was felt that independent validation of the frequency tuning using, e.g. microelectrode recording would be desirable.

Because of space limitations, we did not give sufficient details of the fUS methodology in the submitted manuscript. We agree with you and with reviewers #1, #2 and #4 that this is necessary in order to facilitate adoption of the fUS imaging technique in the future. This is especially true of the new methodological steps we introduced, which enabled us for the first time to attain the fine resolution functional mappings within structures.

We also thank you for the summary of the editorial discussions about our manuscript. While reading your comments and the comments of the reviewers, it became evident to us that we did not emphasize enough the strong methodological novelty of our study. These methodological explanations are important as they are key to achieving the finer resolution that enables fUS for the first time to perform fine functional mapping. In fact, all state-of-the-art applications described in previous publications rely on quite long and basic stimuli (32s in Mace et al., 2011, 30 seconds long visual stimuli repeated 5 times for coarse functional imaging of the visual system in Gesnik et al., 2016, 15 s long stimulus repeated several times in the publication of Osmanski et al., 2015, 10s olfactive stimulus repeated 10 times in Osmanski et al., 2014, large stimulation of the sciatic nerve in Errico et al., 2015) or a single stimulus (Urban et al., 2015, 2s visual stimulus). Using those former methodological approaches, it would have been impossible to reach our goal as it would have required days to perform a 3D tonotopic mapping in the ferret!

Here we pushed the limits of fUS sensitivity to perform very short auditory activations (3s of sound, 8s silence with a random presentation of 5 different frequency tones). With this methodological approach, each high resolution tonotopic slice was acquired in ~15 minutes, thus allowing us to map in 3D the whole auditory cortex and subcortical regions within a few hours, compatible with in vivoimaging. In order to improve the manuscript and clarify better the arguments presented, we highlighted this point in the revised version of the manuscript.

In summary, we have now added significant amount of detail in this revised version of the manuscript to remedy this and other shortcomings. We also provide responses to the other points you bring up such as map variability across animals, comparisons to previous studies, and the time course over which the signals were obtained.

Reviewer #1:The authors use Functional Ultrasound (fUS) to image the auditory brain of awake ferrets with milliseconds temporal resolution and 100 µm spatial resolution. fUS is based on ultrafast Doppler but using ultrasonic plane wave emissions claiming a 50 fold enhanced sensitivity to blood volume changes. Results in three animals (one imaging both hemispheres) show tonotopy in the auditory cortex (MEG, PEG), auditory thalamus (MGB), inferior colliculus (IC) and lateral lemniscus (LL). They also show blood volume changes in the auditory cortex following electrical stimulation of the frontal cortex in a single experiment.Imaging of tonotopic arrangement of different auditory structures seems convincing, whereas the activation of the PSSC/insula by electrical stimulation of the frontal cortex looks vague. Details about how many cases, what place in the frontal cortex and electrical settings were stimulated most effectively must be added.

We have added more details in the revised manuscript (see details below).

We note here that the stimulation experiments were performed in one ferret, and each of the four experiments presented (Figure 3 and Figure 3—figure supplement 1, Figure 3—figure supplement 2 and Figure 3—figure supplement 3) was done one time, on different days. We added this sentence to the revised manuscript to make it clearer (subsection “Frontal cortex stimulation”).

The effect of electric stimulation of frontal cortex in the Claustrum needs to be reported. It has not been reported when the connection is clear (Figure 3C), but not a change in blood volume unless the increase observed in Figure 3B includes both PSSC and Claustrum together. In fact, the increase is observed also dorsal to the PSSC and a mix of increase and decrease deeper than that (Figure 3—figure supplement 1).

The reviewer is correct in her comments. Figure 3C indeed shows clear projections from the frontal cortex to the claustrum, which is located just beneath the insular cortex of the pseudosylvian sulcus (PSSC/insula). It can be hard to delineate structures precisely on US images and we hope that the publication of a ferret brain atlas as well as an assessment of the ferret brain interindividual variability will remedy these uncertainties in the future. We cannot exclude the possibility that our PSSC/insula ROI here encapsulates part or all of the claustrum. However, this does not change the main result of the experiment, which is to show that the upper part of the PSSC/insula is clearly responsive, thus demonstrating that this connection exists. Furthermore, our best hint here is indeed what the reviewer pointed out: when looking carefully at the functional images, one can separate two regions in the depth of the sulcus (illustrated in Author response image 1). We can isolate a pure PSSC area (orange ROI in Author response image 1 panel a), just beneath the large vessel running in the fundus of the sulcus, and a deeper, segregated structure that could anatomically correspond to the claustrum (blue ROI). This region is showing a clear increase in CBV followed by a large decrease (panel b), indicating different dynamics of evoked response in these two spatially distinct areas. This difference in response shape together with the fact that this second (blue) ROI is situated almost exactly where one would expect the claustrum to be, suggests that the claustrum is located below the PSSC/insula in our images. However, since this hypothesis couldn’t be proven with our experimental paradigm (this region is hard to reach, and this spatial and functional segregation was not always clearly visible), we now instead mention the fact that histology revealed FC projections into the Claustrum (Results and Discussion section).

Whether a change in blood volume after far away electrical stimulation is indicative of connectivity or otherwise need to be clarified.

We think that the reviewer is referring here to Figure 3B, where one can observe significant activation, even at positions far from the “optimal” spot of activation (e.g., positions 0 and 3mm). These stimulation areas induced a reduced response with respect to the maximal response obtained at 1.5mm. Our interpretation is that the effect of electric stimulation laterally extends up to 1mm from the electrode tip, because of intracortical connectivity or electrical spreading within FC. Note that this is not the case for comparable distances across depths (Figure 3—figure supplement 1), since close-to-saturation activation appears within ~200µm (from 1 to 1.2mm deep). Future experiments with smaller stimulation currents and a matrix of stimulation electrodes will be necessary to map extensively the pattern of connectivity from FC as a whole to auditory cortex.

Please state where exactly was the stimulating electrode; to say that the evoked activity in the PSSC/insula was maximal for a certain depth and position of the stimulating electrode is imprecise.

We did not emphasize this enough in the main text, due to space limitation. However, in the Materials and methods section, we did state: “[…] electrodes (impedance 200-400kOhms, FHC) were positioned in the FC using stereotaxic coordinates, obtained from functional recordings in behaving animals (AP: 25.5-28.5mm (0 to 3 mm on Figure 2d) from caudal crest / ML: 2mm [14])”. The position of the electrode in terms of Antero-Posterior position is 25.5+1.5 (max. in Figure 3B) = 27mm from caudal crest, and 2mm lateral from the middle crest. We have now added a sentence to explain how we determined the antero-posterior position of the caudal crest (5mm lateral from the medial crest, as suggested in the ‘in press’ ferret brain atlas Cyto- and myeloarchitectural brain atlas of the ferret (Mustela putorius) in MRI aided stereotaxic coordinates, Radtke-Schuller, 2018,). These coordinates were chosen based on physiology experiments showing that this region was a good candidate for top-down modulation of auditory cortex [Fritz et al., 2010]. We now make explicit that we targeted the region in between the anterior part of the anterior sigmoid gyrus (ASG) and the posterior part of the proreal gyrus (PRG) during this experiment (subsection “Frontal cortex stimulation”).

In terms of depth, we cannot be certain of the exact stimulation depth within ASG/PRG because of scar tissue regrowth on top of the brain (hence the statement in the caption of Figure 3—figure supplement 1: “Here, 0 was set at the surface of the tissue covering the brain, which can be up to 1mm thick”). During electrophysiological experiments in similar craniotomies, we often detect spiking activity starting from about 0.5 to 1.0 mm from tissue surface, thus suggesting that the surface of the brain starts within this range. This interpretation suggests that our optimal spot for stimulation is between 0.5 and 1mm below brain surface (~1.5mm from tissue surface).

The decrease of blood volume in MEG is indicating some kind of polysynaptic inhibition?

In interpreting our results, we were partly relying on a previous fMRI study from the Logothetis’ lab (Logothetis et al., 2010) which showed that electric stimulation in the LGN elicited BOLD increase in V1 but decrease in extrastriate cortical regions (in particular V2 and MT). When they injected a GABA antagonist in V1, they observed a reversal of the effect, namely that the decrease of BOLD in the extrastriate regions turned to an increase. This indicates that LGN electric stimulation may have induced a substantial recruitment of the inhibitory network in V1, shutting down the local excitatory network. As a consequence, V1 excitatory output activity decreased, correlating with a diminishment of BOLD signal in the downstream cortical regions.

Therefore, if one assumes that this finding is relevant for fUS imaging, then the observed decrease in CBV could be due to excitatory polysynaptic connections between the PSSC/insula and the MEG. In fact, based on previous anatomical work in the ferret (Figure 9 in Bizley et al., 2015), such indirect connection between PSSC/insula and MEG may be mediated by AEG. We therefore cannot exclude the possibility that this decrease in MEG activity is due to subcortical or out-of-plane structures that we were not visualizing.

If PSSC/insula area is not driven by sensory stimulation (broadband noise or visual stimulation), is suggested that is not a sensory structure?

The plane that is visualized in Figure 3 and Figure 3—figure supplement 3 corresponds to the PSSC/insula, i.e. the part of the PSSC below the fundus of the sulcus. It actually shows a significant response to broadband noise in the PSSC/insula (Figure 3—figure supplement 3), but not to visual flashes, consistent with Manger et al., 2005. As stated in the text, this is a weak response compared to A1 (~5% instead of 15%). It is actually difficult to relate the sensory modality selectivity of the PSSC/insula to previous works, as its location just beneath the fundus of the PSS makes it difficult to reach with classical electrophysiological recordings. As far as we know, previous anatomical characterization of the PSSC focused on its anterior (anterior ectosylvian sulcal field in Bizley et al., 2007, anterior PSSC in Bizley et al., 2015, anterior ectosylvian visual area, Manger et al., 2005) and posterior banks (posterior PSSC in Bizley et al., 2015), and not on its insular part. Additionally, we note that other auditory stimuli may be able to elicit larger activity in PSSC/insula (such as more complex stimuli, ferret vocalization, etc.) and further research would be essential to go beyond this proof-of-concept.

Comparison in tonotopy is tricky across cases. Figure 1 is the best example but high frequencies in MEG are not only in the tip of the gyrus but also in between A1 and AAF. The other two cases were very different with few high frequencies in V_right_ and AAF full of high frequencies in Bright case (Figure 1—figure supplement 2). Also, in these two cases the tonotopy in IC and DNLL look less convincing than in Figure 1. The authors should explain in detail the different cases and discuss the differences between cases.

The reviewer is correct in pointing out the variability between the maps in the different animals and structures. We expected such variability, however, based on previously published maps (Bizley et al., 2005) and our own experience in ferrets where inter-individual differences was the rule rather than the exception. Indeed, in the five animal’s maps illustrated in Bizley’s Figure 4, one can see animals with a high frequency region for both A1 and AAF all along their common border (panel D, ferret F0333) or with the high-frequency region of AAF along the same border (panel E, ferret F0321). We have now added text to the discussion of this point in the revised manuscript (Results and Discussion section), in the limits of available space. We can develop our discussion in the revised manuscript if the editor allows us to do so.

The paper will improve if further details are added.

We thank the reviewer for suggesting these improvements and have now added and discussed all the necessary details more extensively in the revised manuscript (see points below). This also includes schematics of the acoustic or electrical stimulations in Figure 1 and Figure 3 and detailed explanations of the stimulation protocol in the Materials and methods section.

For example, how many repetitive days of imaging is not clear until Figure 1—figure supplement 3 (even here it is difficult to read the y-axes legend).

Tonotopic maps (e.g., Figure 1) are usually recorded within the same day. Each of the maps shown in the manuscript was recorded on a different day. The only experiment where we recorded the same slice over days is depicted in Figure 1—figure supplement 3 and Figure 2—figure supplement 2. This experiment was performed in order to assess the stability of the recordings. Figure 1—figure supplement 3 shows that even after removing the probe, putting the animal back in the cage and setting it up again the next day, we were able to minimize the repositioning error on the order of the PSF width (error of ~200µm here) (before any numerical co-registration) when imaging the next day. Figure 2—figure supplement 2 shows that the tonotopy is relatively stable over days. Furthermore, all the improvements shown in Figure 1 are doable in single slices.

Where exactly was the electric stimulation in the frontal cortex was applied: anterior sigmoid gyrus, lateral part, at what depth?

This comment is addressed in our response above.

LL is related to lateral lemniscus fibres, lateral lemniscus nuclei, only the dorsal nucleus of the lateral lemniscus as suggested by one of the figures.

US images allows one to coarsely identify structure, but it is still a hard task to determine which subparts of these structures are observed. Here, the function can help. From the literature, only the dorsal nucleus of the lateral lemniscus (DNLL) shows a clear tonotopic organization (in the cat, Bajo et al., 1999, Malmierca et al., 1998). Based on this and on comparison with the ferret brain atlas, we therefore think that the tonotopically organized structure that we were able to observe, ventral to the IC, is likely to be the DNLL. However, we preferred only to identify it as the ‘lateral lemniscus’, in order not to overstate our description.

The patterns of sensory and electrical stimulation are not clear; 2 s stimulation every (8 + 10 s) of silence or only 8 s of silence? In this 2 seconds of stimulation, how long is the stimulus?

We apologize for the confusion. In the Materials and methods section, we describe:

“The protocol for sound presentation is as follows: 10 seconds of silence (baseline), then 2 seconds of sound followed by 8 seconds of silence (return to baseline).”, where the sound is a pure tone presented during these 3 seconds (not 2s, there was a typo that we corrected in the revised manuscript). This defines a trial; and trials follow each other with only a little random jitter in time of about 1 to 3s. fUS acquisitions are synchronized with the beginning of each trial.

We note here also that for Figure 1—figure supplement 3 and Figure 2—figure supplement 2 we used an even faster protocol (2s tone, and random interval of 4 to 6s – uniformly distributed – between two tone-presentations) that allowed us to present more stimuli (75 per frequency) in a relatively shorter time (~45 min), as stated in the Materials and methods section.

Elsewhere in Materials and methods section, concerning electrical stimulation protocol, we also stated that:

“Each trial consisted of 10s of baseline, then 6s of monophasic stimulation at 100 Hz and 200µA (2ms pulses, 200ms-long train, repeated at 2Hz), after a return to baseline of 10s”, where trials unfurl in the same way. We have now added a schematic in the respective figures to clarify this point and improve the description in Materials and methods section.

The% CBV as percentage of cerebral blood volume is not explained until subsection “Signal processing, analysis & statistics”.

We introduced% CBV earlier in the revised manuscript, both in the main text (Results and Discussion section) and in subsection “fUS imaging”).

The time resolution of the technique seems to be more in the range of seconds than milliseconds when the quantified voxel responses are taken in a time-window 3-5 seconds after sound onset. Could the authors explain it?

To estimate the local blood volume change, ultrafast imaging provides one image every about 2ms. Thus, for each second, a set of 300 images is acquired (corresponding to a 600 ms acquisition period). We emphasize that this high temporal resolution is highly important because it enables us both to cancel unambiguously respiratory and pulsation artifacts using advanced spatio-temporal filtering (Demene et al., 2015) and to achieve part of the increased blood volume sensitivity. After this spatiotemporal filtering, the complete set of data from the 300 images (sampled at 500 Hz) is averaged to produce one single cerebral blood volume image, leading to a temporal resolution of 1 second for functional imaging. We deliberately chose this averaging process because (as the reviewer correctly points out) we don’t need to use the temporal resolution at its maximal value since the physiological response is slow, being driven by the neurovascular coupling, and we prefer to increase in this way the signal to noise ratio.

We have now improved the text in Materials and methods section in order to clarify this point:

“Although the ultrafast 2ms temporal resolution is available for the CBV image generation, they are in fact averaged into one CBV image every second to capture the dynamics of the cerebral blood physiological response. Nevertheless, it should be noted that this rapid sampling rate is a key asset to unambiguously cancel any respiratory or tissue pulsatility artifacts in the final averaged images”.

Red and blue color code in figures is extremely confused. It can apply to increase and decrease in blood volume, to auditory cortex and visual cortex, or even high frequency and low frequency of auditory stimulation. I suggest the use of different colors when possible and always add a color code scale to each panel.

Excellent advice. This is now modified accordingly in all relevant figures (Figure 1—figure supplement 1 and Figure 3—figure supplement 3), and we added clearer colorbars for the tonotopy (Figure 1 and Figure 1—figure supplement 2 and Figure 2—figure supplement 2).

The decoder accuracy is very poor, at least in the auditory cortex and MGB (Figure 2). It would be good to add a plausible explanation about the fact that is better in the middle layers of the cortex (about 500 microns).

It is not straightforward to assess directly whether a ~55% accuracy is adequate or not because this estimate depends on many factors: the number of recorded trials, number of voxels, number of stimuli (here 5, bringing a 20% chance accuracy). Here our goal was to demonstrate that we can get single-trial decoding accuracy with only 9 trials of each frequency in the training set, and the resulting performance is clearly above chance level. Moreover, note that the type of classifier that we used here was a simple linear decoder, and that the fMRI community has been expanding the number of tools available over the past 20 years, including more sophisticated decoders for BOLD signal. Further developments in the lab will focus on adapting these tools to fUS signals.

Previous studies (Guo et al., 2012) showed that neurometric sensitivity in pure tone classification was better in middle cortical layers, compared to superficial or deep layers. This is in line with our finding, and we therefore edited the manuscript to make this interpretation clearer (Results and Discussion section).

This technique could be a great complement to another imaging and recording techniques used in parallel with behavior in awake preparation. It would add value to include a new section of future potential applications where limitations were also discussed, for example having the head fixed or the need for sedation.

We thank the reviewer for this comment. This is definitely the next step we are heading towards. We added a discussion on this topic (Results and Discussion section), in the limits of the available space. We can develop our discussion in the manuscript if the editor allows us to do so.

Reviewer #2:1) As the key novelty of this paper critically relies on the multivoxel pattern analysis/decoding accuracy and discriminability/resolution index calculations, these analysis procedures need to be described much more carefully and the numerous ad hoc thresholds currently described in the analysis section appropriately justified. How the comparison for multiple corrections is done also needs clarifying. If this work is meant to speak to the neuroimaging community at large, it needs to employ standard neuroimaging analysis as well as provide a link between its output and the metrics currently employed.

The MVPA analysis did not employ any thresholding, explaining why there is no description of such a procedure in subsection “Decoding”. Thresholding was only used for visualization purposes (for instance in Figure 1), but all analyses shown in Figure 2 were based on the ensemble of voxels recorded in auditory cortex, be they tuned or not. This is now stated explicitly in subsection “Decoding”: “No thresholding procedure was used in this analysis.”

Moreover, the use of a randomization procedure takes into account the correction for multiple tests.

Please see response to reviewer 1 concerning the relationship of our analyses to the standard neuroimaging methods. Briefly, the classification procedure we used here was the simplest possible (linear decoder without any selection procedure, except ROI selection), so as not to obscure the data, and also leaving room for increasing complexity of such a decoder later. Further development will focus on improving these analytical methods (especially MVPA) and adapt it to fUS signal, inspired mainly by the huge and still-expanding fMRI analytical framework.

2) Given that both spatial coverage and resolution reported critically depend on the extensive skull removal, the limitations of this technique as a means of studying across-region connectivity needs to be carefully qualified. This also makes current comments on how the technique compares to fMRI resolution ill grounded: if invasiveness is allowed, then implantation (even on pial surface!) of RF coils also affords much higher spatial resolution than what the authors currently quote.

We fully agree with the reviewer that our comparison was somewhat clumsy. We have now added this comment in the revised manuscript to point out this drawback:

“Relative to fMRI, it has much faster acquisition rates (ms rather than seconds) leading to a fine discrimination between blood flow and motion artifacts (breathing motion, tissue pulsatility, …), substantially higher spatial resolution for cerebral blood flow imaging at the expense of non-invasiveness, greater portability and lower cost, and versatility for awake animal imaging.”

To clarify further to the reviewer the above assertion, we would like to mention that fUS is capable of transcranial imaging, and that there is a trade-off between skull penetration and spatial resolution. With the 15 MHz frequency used in our study to achieve 100 µm resolution, the thick skull of the ferret had to be removed. But decreasing the ultrasonic frequency would have enabled us to perform transcranial imaging at a lower resolution. In mice, where the skull is much thinner, fUS imaging can be performed non-invasively at 15 MHz (Tiran et al., 2017) with 100µm pixel resolution, a higher resolution than in non invasive fMRI for mice.

3) The rigor of figure creation is subpar: all of the images that have colored overlays need to have a color bar alongside with numbers reflecting the mapping of those colors to actual values of physiologically interpretable quantities. The reported CBV changes are very high compared to the literature, even in the awake mammals, so a discussion on this topic is warranted as well.

All figures have been modified as per the suggestions of the reviewers. We also added a section to discuss the high amplitude of CBV changes (Results and Discussion section). Interestingly, responses in non-human primates are even larger (personal communication, Pierre Pouget, ICM, Paris, France). fUS imaging has also been performed in human neonates, and showed large variations of UfD (Ultrafast Doppler) signal (proportional to CBV signal): for example, up to +/-50% during active sleep, and up to +80% during epileptic seizures (Demene et al., 2017).

Reviewer #3:The paper titled "Multi-scale mapping along the auditory hierarchy using high-resolution functional UltraSound in the awake ferret" – submitted as a Tools and Resources article in eLife – reports imaging of tonotopic maps in the awake ferret and imaging of activity evoked in auditory areas by electrical stimulation of the frontal cortex in the sedated ferret, with functional ultrasound imaging. The authors present these results as a new method for recording brain activity at multiple spatial scales at higher resolution in awake animals, broadly applicable for other neuroscience questions. The imaging of tonotopic maps could be of interest for the auditory field, particularly in deeper structures difficult to access. However, there are major problems concerning both the novelty and the relevance of these experiments to justify the claim made by the authors in term of methodological advances. For this reason, I do not recommend the publication of this article as a Tool and Resources article in eLife.

We respectfully disagree with these comments about the novelty of our methodology. We respond here in detail to each of the reviewer’s assertions. We, however, agree that we did not emphasize enough the methodological novelty of our experiments in relation to previous efforts. These methodological explanations are of course important as they are the key to reaching the finer resolution that enables fUS for the first time to provide fine functional mapping within structures.

As mentioned above in response to the editor’s summary, all state-of-the-art applications described in previous publications rely either on the repetition of a single stimulus (Urban et al. 2015) or on long lasting stimuli (32s in Mace et al., 2011, 30 seconds long visual stimuli repeated 5 times for coarse functional imaging of the visual system in Gesnik et al., 2016, 15 s long stimulus repeated several times in the publication of Osmanski et al., 2015, 10s olfactive stimulus repeated 10 times in Osmanski et al., 2014, large stimulation of the sciatic nerve in Errico et al., 2015). Using those former methodological approaches, it would have been impossible to reach our goal as it would have required days to perform a 3D tonotopic mapping in the ferret!

Here we have pushed the limits of fUS sensitivity to perform rapidly short and diverse auditory activations (3s of sound, 8 s silence with a random presentation of 5 different frequency tones). With this methodological approach, each high resolution tonotopic slice was acquired in ~15 minutes, thus allowing us to map in 3D the whole auditory cortex and subcortical regions within a few hours, compatible with *in vivo* imaging. In order to improve the manuscript and clarify better the arguments presented, we highlighted this point in the revised version of the manuscript.

Major comments:1) There is no significant technological or methodological advance compared to previous work. The authors claim that they did for the first time (a) in an "awake" animals (b) "multi-scale mapping" (c) at "high-resolution". These three points have been shown before with the same technique.

We disagree with this comment. For the sake of clarity, we begin this response by citing the full sentence that the reviewer was likely referring to:

“Here we show for the first time its capability to resolve the functional organization of sensory systems at multiple scales in awake animals, both within structures by precisely mapping sensory responses, and between structures by elucidating the connectivity scheme of top-down projections.”

a) There were three papers describing functional ultrasound imaging, including two papers in awake freely-moving rats (Macé et al., 2011; Urban et al., 2015 and Sieu et al., 2015) (one was not cited by the authors).

We did not in any way intend to claim that we are the first to record in the awake animal. We would not do this, since we cited studies employing the imaging technology in freely moving rat! Instead, we had emphasized the novelty of our results compared to previous studies with respect to functional high-resolution tonotopic mapping, both in cortical and deep structures at this level of fine spatial resolution. We also demonstrated that the functional mapping reaches a spatial resolution comparable to the ultrasonic spatial resolution. Finally, we noted that the true spatial resolution of functional ultrasound imaging was never demonstrated before in physiological responses, which is of major importance.

b) The possibility to record at "multiple spatial scales*" (defined by the authors as the possibility to image mesoscale patterns of activity within a brain region and across brain regions) has been shown previously. For example, a single barrel was mapped in the rat brain, as well as large scale activity during epileptic seizures (Macé et al., 2011). Odor topographic maps have been imaged within the rat olfactory bulb (different glomeruli) and piriform cortex (Osmanski et al., 2014). Large-scale connectivity was explored previously in the awake rat (Osmanski et al., 2014). Imaging of sensory responses in small deep subcortical nuclei were shown in freely moving rats (Urban et al., 2015). A recent paper by the authors reported mapping of the visual responses in the rat brain at large-scale as well as local scale (for different retinotopic positions) in visual cortex (Gesnik et al., 2013).*

As far as we know, nobody has done functional mapping of physiological responses at the resolution we describe with functional ultrasound imaging, nor indeed of functional responses at different layers of the cortex. We should know this given that the reviewer cites our own publications to make his point.

Thus, Macé et al., 201 did not describe a method to provide maps of barrel cortex somatotopy. Instead, whiskers were successively cut leaving only a single one; further, we did not demonstrate that the extension of the activated area was a single barrel. In reference (Osmanski et al., 2014), the whole piriform cortex was activated for different odors and the activation maps in the olfactory bulb were not able to localize glomeruli but only deep vs near surface activations. (Osmanski et al., 2014) and (Osmanski et al., 2014, not on awake animals by the way), also deal with the spatial extent of activated regions and connectivity mapping, both are very coarse. Finally, in our most recent paper cited in this comment (Gesnik et al., 2017) there was no retinotopic mapping of any kind presented in the paper because we were not able to reach the sensitivity and thus, the required resolution to provide retinotopic maps of activation. Just as in other publications, reference (Gesnik et al., 2017) used coarse activations, specifically light flashes in the right or left hemifields to characterize differences in the right and left visual cortices. We have pointed out this limitation of the results in the discussion of reference (Gesnik et al., 2017) and highlighted the need of a proper evaluation of the detection limit of fUS in order to be able to achieve a proper estimate of the fine spatial analysis for a real retinotopic mapping. For example, we said: “Furthermore, the high in-plane resolution of fUS could be leveraged to more finely segregate areas of the brain according to their function. For instance, reducing the width of the stimulation sectors on the screen, would yield the detection limit of fUS in term of spatial extent of the responding cerebral areas.” (Quote from Gesnik et al., 2017.)

Our current study of the tonotopic organization precisely addresses and answers these questions by “zooming in” within each brain structure and focusing on a completely different functional scale in small and deep structures.

So, we respectfully repeat again, that we are fully aware of all the coarse-grained recordings in the olfactory and in the different regions of the barrel cortices. They are definitely at a different order of resolution to anything we describe here.

c) The resolution of the technique (100 µm) has been demonstrated experimentally (point spread function) and theoretically in a previous paper (Macé et al., 2013). Therefore, the title "high-resolution functional ultrasound", as well as claims about spatial resolution made in the article are misleading, given that no improvement was made. In fact, the ultrasound acquisition and data processing used in this work is the same as in previous papers (for example in (Gesnik et al., 2017)).

Once again, we disagree with this assessment by the reviewer. In our previous paper (Macé et al., 2013) and other papers on fUS imaging, it has been demonstrated that the spatial resolution of the Doppler ultrasound maps is 100 µm. But it was never demonstrated that two neighboring pixels were able to carry independent physiological information during activation. This is ultimately the resolution of functional ultrasound. What is demonstrated in (Macé et al., 2013) is the resolution of CBV maps provided by ultrafast Doppler. In this previous publication there was no physiology, no sensory responses, and no quantification of the functional resolution for stimulus-evoked physiological responses. This is not at all the kind of high-resolution imaging we describe in this paper using sensory responses. We already explained this point in the introduction of the manuscript, but we emphasize more in the revised version: “Also, if the theoretical spatial resolution of Ultrafast Doppler for high sensitivity mapping of cerebral blood volume (CBV) changes has been shown to be 100 µm for whole brain imaging in rats 2, the ability of the fUS technique to measure independent information on functional brain activity at such a small scale, i.e. the truly informative fUS imaging resolution, remains to date unproven.”

d) Concerning the temporal resolution, the authors claim in the introduction that functional ultrasound imaging has "ms resolution compared to fMRI" (Introduction) and use the term "rapid" multiple times in the manuscript. However, they acquire one image of one brain slice in 1 second (subsection “fUS imaging”), and the whole tonotopic map in several hours (subsection “Protocol for sensory response acquisition”). Although the spatial resolution of fUS is better than fMRI, this is not the case for the temporal resolution.

There is a misunderstanding here, as we never said that fUS has a ms-resolution (actually the quote does not come from the paper (does not show up in the Introduction)). It is obvious that hemodynamic responses cannot provide this temporal resolution. What we said (once) is that the technique has “temporal dynamics” of that order compared to the (seconds) in fMRI. That is completely different. It refers to the rate at which we can take the images. This is a key feature in fUS as it enhances the sensitivity significantly compared to the fMRI, which at best can take images over seconds. This very high temporal sampling allows us to filter out respiratory and pulsatility artifacts with a very high precision, thus enhancing the S/N. Therefore, single-trial measurements are easy to do in fUS, but are difficult in fMRI because of their poorer S/N. We also used the word rapid always to compare to other mapping techniques such as microelectrode mappings. In that sense, taking hours to get all the maps shown from one animal is truly rapid compared to the days and months it takes to get comparable maps with a different technique. We have further clarified these points in the revised manuscript.

In the Materials and methods section, we have added a sentence to point out the importance of high temporal resolution of ultrasonic imaging for artifacts rejection and recall again our 1s resolution for fUS imaging: “Although the ultrafast 2ms temporal resolution is available for the CBV image generation, they are in fact averaged into one CBV image every second to capture the dynamics of the cerebral blood physiological response. Nevertheless, it should be noted that this rapid sampling rate is a key asset to unambiguously cancel any respiratory or tissue pulsatility artifacts in the final averaged images”.

2) Beyond the problem of novelty, there is a major issue with the claim about the "awake" state in this paper. The second part of the paper, about long-distance connectivity, was done in sedated ferret (subsection “FC stimulation”). This is not clear to the reader except in the Materials and Method section. The paragraph on connectivity experiments starts with: (Results and Discussion section: "Localizing and quantifying such connection in awake animals, remains technically challenging […] Here we show that fUS can be used to probe functional connectivity between two brain structures") leading the reader to think this is done awake. The choice of this experiment, in the context of demonstrating the interest of a new tool for imaging awake ferrets, is disputable. Whereas only tonotopic maps were recorded in awake (restrained) animals, the awake state is emphasized at multiple instances, including in the title and abstract. Moreover, obtaining tonotopic maps do not require awake state (some examples under anesthesia: (Nelken et al., 2008; Bizley., 2005 and Mrsic-Flogel, Versnel and King 2006)). A key advantage of functional ultrasound is to be applicable to awake, behaving and freely-moving animals, as shown previously (Urban et al., 2015; Sieu et al., 2015). Although many interesting biological insights can be obtained under sedation, this limits the interest of this work compared what was previously published, in particular for a methodology paper.3) As an addition to the point 2, the authors report in the Materials and Method section that "the ferret was sedated to avoid movement artefacts". Previous studies were done in freely moving animals without motion problems (Urban et al., 2015; Sieu et al., 2015). Why is motion a problem in connectivity experiments compared to tonotopic experiments is unclear and not discussed. The claim in the conclusion that this method is "readily adapted to mobile and highly stable configurations" is undermined by the fact that the connectivity study was done in sedated animals.

We clearly dispute the reviewer’s erroneous implications in this comment. The sedated state of the animal was explicitly stated twice in the main text (Results and Discussion section) and in the Materials and methods section. More importantly, however, we want to stress that sedation is not anesthesia. We used a tiny proportion (15%) of the full dose of sedative just to relax the animal. The animal was certainly not anesthetized. The reason for sedation was twofold: firstly, ethical and humane consideration associated with the potential side-effects of electric stimulation. Secondly, there were technical limitations as the probe in all our experiments was not directly fixed onto the animal implant. The reason for this was simply because in order to perform all the experiments shown here, we had to move and rotate (all angles) the probe according to the region of interest under focus. Thus, it was much easier in our case not to attach the probe to the skull. Furthermore, this configuration with the head fixed was also beneficial for reproducible auditory stimulation as the earphones could be always adjusted in the same fashion. However, future developments in our lab will allow us to use the fUS with an attached probe in the ferret.

In clinical neuroimaging, slight sedation is also used for neonates fMRI imaging (Counsell and Rutherford, 2002; Erberich et al., (2006); Isobe et al., (2001)). As explained in Tocchio et al., “Beyond age 3 months sedation is almost always required to achieve a good quality motion free study”. Whereas slight sedation is commonly accepted for newborns fMRI, anesthesia is highly prohibited for ethical considerations (Tocchio et al., 2015). Sedation does not mean anesthesia.

In summary, “slightly sedated” is just that, it is definitely not “anesthetized” and few in the neuroscience community would dispute this.

4) Functional ultrasound imaging technique is applied here in a different animal model, the ferret. However little effort is done to provide readers with a protocol for reproducing the work. For example, identification of brain regions is an important step for other users with different questions. Identifying brain regions anatomically on ultrasound images is not easy compared to, for example, MRI. This step is not clearly explained in the article or in the Materials and Method section. It was apparently done manually and/or based on the auditory responses (which is not translatable to other behaviors). The usability for other neuroscientists to study other questions in the ferret, for example to reveal new regions implicated in a given behavior, is therefore limited. Availability of the ultrasound codes for other users is not stated.

Unlike other species (rodents, primates), there was not yet a ferret brain atlas publicly available. This makes it difficult to give coordinates beyond what is already described in the paper. However, we did provide anatomical coordinates in the manuscript in the Materials and method section). We now make this description clearer and take this opportunity to mention the upcoming publication of a ferret brain atlas (http://www.springer.com/us/book/9783319766256), confirming the growing importance of the ferret as an animal model in Neuroscience.

Regarding the ultrasound codes, they are all available within the framework of research collaboration agreements between academic institutions as is usual. We have added a reference to this in the manuscript.

5) The connectivity experiment, used to demonstrate top-down projections from the frontal cortex to the auditory system, is conceptually problematic. Electrical stimulation is known to evoke antidromic activity; therefore, it is not possible to discriminate between top-down and feed-forward connections if reciprocal connections exist. Optogenetic activation would have helped resolving this issue (and would have broaden the applicability for other neuroscience questions). Compared to simple tracer injections, it is unclear what we learned from this experiment, in part because the functional data presented are not discussed and all the focus is put on the method.

The reviewer’s point about a possible contribution of antidromic activity cannot be excluded; we now emphasize this clearly in the manuscript (Results and Discussion section). In addition, we also discuss in detail the putative polysynaptic connection between FC and MEG (see reviewer 1’s comments). Last, we agree that optogenetic stimulation would be a more elegant approach than electric stimulation. However, since it is somewhat more difficult to conduct in ferrets (compared to the mouse), we opted for the electric stimulation at this stage of the project, but we definitely agree that it is the way to go in subsequent experiments. We added a section on this in the Results and Discussion section.

Comments on Methodology:- Tonotopic maps (Materials and methods section). The authors display voxels in tonotopic maps if the max response is above a certain threshold (15%) and if the max response is correlated with the average hemodynamic response function (p<1e-3). Then they indicate: "these thresholds are adjusted for different structures and different animals". First, the same thresholding parameters should be used for the three structures of the three animals. Second, the frequency tuning is not taken into account in this thresholding method, only amplitude. Any voxel responsive to sound, regardless of whether it is tuned or not, would pass this threshold. A statistical test should be used on each voxel to determine significant tuning, and only tuned voxels should appear on the maps. Third, the authors applied a mask manually to show only voxels in auditory structures (subsection “Signal processing, analysis & statistics”). This biases the results and is misleading. It is not possible to assess the quality of their thresholding method outside of the auditory regions.

First of all, we emphasize again here that thresholding procedures were only used for visualization purposes. As pointed out to reviewer 2 (above), all the quantitative analysis (MVPA and functional resolution quantification) was performed on non-thresholded data.

Second, we used a threshold based primarily on sound responsiveness and not tuning in order to visualize also the regions that did not display any tonotopic organization, in order not to bias our interpretation and to be able to detect any potential auditory structure (for example, AEG that shows a poor tonotopic organization (Bizley et al., 2005), or a structure just anterior to the visual LGN that shows responses to pure tone and isn’t organized tonotopically). The tonotopic organization here emerges where it is present. What we mean is that voxels that do not reach significance at, for instance, 5% (which is likely since we play only a few trials) can be consistent and make sense overall, as a spatial map, and should thus be preserved when making a qualitative assessment (whereas overall responsiveness takes into account more trials and is thus less conservative).

Third, overall responsiveness between animals differs, even in the context of similar experimental conditions. We did not find any consistent interpretation for this variability, except biological differences in the hemodynamic responses of the animals (as discussed several times here). Thus, different thresholds for different animals can be justified for visualization purposes.

Finally, applying masks can be justified when our goal is to display only regions of interest, of which proper visualization can be impeded when figures are crowded, for example by overall noise in the recordings (which is unavoidable and visually cumulative when many slices are superimposed) or other structures (we have mentioned before, for example, a structure just anterior of LGN).

Please find, however, in Figure 1 where thresholds were obtained by computing the ANOVA p-value across frequency per voxel (threshold at 10%, not to be too conservative considering the small number of trials) and masks have been removed. We note here that we fixed a small bug affecting the display. As one can note, this criterium is more conservative and some regions are not shown anymore (such as AEG), but the overall results are fundamentally the same. We can modify this Figure and the related others in the revised manuscript if the Editor deems it necessary.

- Significance of the frequency tuning is tested only for one single voxel of one animal (Figure 1C, subsection “Signal processing, analysis & statistics”). This is uninformative. Average tuning with respect to preferred frequency of all voxels would have been the right analysis to do. This should be done for all regions separately, compared to control regions of the same size and compared across animals.

Figure 1C was intended as a simple example pixel, to give the reader a sense of the responses and tuning curves. We are not sure we understand the purpose of the analysis proposed by the reviewer. For instance, grouping voxels with random tuning curves by their maximal response frequency will always provide tuned average tuning curves. Therefore, we think that MVPA is the proper (and standard) analysis to show in a simple and concise way that the information is there (and is fully controlled using cross-validation and randomization procedures).

- Displaying an average tonotopy map for each brain region would be beneficial. The organization in each region is not clear from the different examples. In particular, in Figure 1—figure supplement 2, maps of the third animal are different from the other two animals (for example, in PEG). No comparison is made with the known tonotopic maps from the literature (such as: axis of the tonotopic map, reversals/boundaries of the maps) (Nelken et al., 2008; Bizley., 2005 and Mrsic-Flogel, Versnel and King 2006).

We discuss this particular point in reviewer 1’s section. There is indeed some variability between the different animals, but this variability is expected from the literature (Bizley et al., 2005) and our own experience in ferrets (see response to reviewer 1’s request for more detailed examples). We now discuss this point in the revised manuscript (Results and Discussion section). We also thank the reviewer for pointing out these references that were accidentally omitted; they are now included in the Results and Discussion section.

- The interest of the multivoxel pattern analysis decoding is disputable. The analysis shows that, when pooling all voxels from one structure, single trial responses carry significant information about the frequency tuning. First, that does not mean that a tonotopic map can be reconstructed at the single-voxel level from a single trial. Second, this kind of analysis is usually used to determine if a property (i.e. sound frequency tuning here) is encoded in a brain region when the spatial resolution is too low or if responses are intermingled. Here it is clear that frequency tuning exists in auditory areas. The claim that " the hemodynamic signal imaged in fUS is reliable enough to decode brain activity on a single-trial basis within a single experiment" (Results and Discussion section) is trivial. It could have been demonstrated in a much simpler way – for example by averaging single-trial tuning curves of all voxels of the tonotopic map in a specific region.

First, we did not claim that MVPA was used here to reconstruct ‘tonotopic maps […] at the single-voxel level from a single trial’. Instead, we used MVPA to ‘estimate the reliability and selectivity of fUS single-trial responses’. And indeed, having shown that tuning was present in the auditory cortex, the fact that single-trial brain activity decoding is feasible becomes trivial. The question here is rather to which extent this is doable, with a restricted number of trials and voxels. Note that good MVPA decoding accuracy here is not related to the presence of a tonotopic axis, and we did not claim these two were directly related. What we provide here is a quantification of the information content of the fUS signal. Note also that, actually, using a simple linear decoder as we did is much simpler than selecting regions and averaging the tuning curves, since no manual selection is required in our case (except to focus on a structure) and that averaged tuning curves over voxels selected according to their best frequency will always show a tuning. Finally, this allows us to pave the way towards more sophisticated analysis, closer to the analysis procedures of fMRI (see response to reviewer 1).

- The quantification of the spatial resolution was made based on 6 voxels hand-picked in one animal and one structure (Figure 2B). It is not reported where (in which brain structure? which animal?) and these voxels are not put in the context of a larger map. This is not sufficient to sustain this claim. Such quantification should be made on more voxels, in different structures and compared across the three animals.

Since functional resolution can only be evaluated in a region where there is an abrupt variation in spatial encoding, this explains why we focus on a particular zone. We felt that showing one example is enough to prove our point. A similar rationale was followed in Errico et al., 2015, in which ultrafast ultrasound localization microscopy was used to measure the profile of speed within *two* cortical blood microvessels, and therefore demonstrating the spatial (and not functional) resolution of this ultrasound-based microscopy. The goal of this demonstration thus, is not to be compared across structures. However, to demonstrate further that we were not reporting a unique isolated event, we now provide more examples in the Figure 2—figure supplement 3 (from Bright’s IC (a), and Vleft’s AC (b)). Pixels taken into account for the analysis are the circled ones. We note that this new example relies on only 10 trials, so half of what is presented in Figure 2B but shows a similar conclusion (Mean response resolution 100 µm, and Tuning curve resolution ~200/300 µm). We also replaced Figure 2C with a new and clearer example from V_right_’s AC (10 trials) where resolution is similar, and we put the transition in its context (left panel). We also revised the caption of the related figure where this was taken from; we thank the reviewer for pointing this out. We put S_right_’s__ AC in Figure 2—figure supplement 3 in (c). Thus, functional resolution was similar across animals, and across structures.

- Statements about the multimodal responsiveness of brain areas (Results and Discussion section, Figure 3—figure supplement 3) are not sufficiently supported. The authors report the data obtained from only one animal. Generalization on the multimodal aspect of these regions would require a statistical test across different animals. Moreover, as mentioned before, the anatomical identification of brain regions should be standardized based on an atlas and not manually picked after the functional recordings. For example: responses in PSSC seem restricted to the upper voxels of the delimited region (Figure 3—figure supplement 3). This could be due to a misalignment of the region boundaries, thus misattributing voxels from the neighboring auditory responsive regions to PSSC.

We agree that this finding would need further evidence, explaining why we chose to integrate it as a Supplementary figure. Nevertheless, we want to highlight the fact that we anatomically captured the PSSC/insula, as the radial organization of the blood vessels in this structure clearly allowed us to delineate the fundus of the PSS. Previous physiological investigations also found auditory responses in the PSSC/insula (Manger et al., 2005) and anatomical studies showed projections from the AEG to the insula (Bizley et al., 2007 and 2015). These findings are in line with the auditory responses that we observed in the PSSC/insula, and we therefore doubt that we did not correctly delineate the PSSC/insula. More generally, one can combine Figure 3 and Figure 3—figure supplement 3 and see that electrical stimulation evoked activity overlaps with sound evoked activity, independently of any ‘manual’ delimitation. This region, located just beneath the blood vessel in the fundus, corresponds undoubtedly to the PSSC/insula.

Reviewer #4:Overall the study looks impressive and interesting to me, and I have just a few statistical questions.

We thank the reviewer for these positive and encouraging comments.

The primary voxelwise activation response mapping was thresholded both in terms of minimal% signal change and statistically. However, the latter does not appear to have corrected for multiple comparisons across voxels and frequencies. This needs to be done for clarity and transparency. However, I'm not saying that any potential lack of full family-wise-error significance in the initial voxelwise mapping would necessarily cause concern for the later multivariate analyses or overall story.

We did not use multiple comparison correction for the displayed images. Indeed, we think that the tonotopic organization we find, consistent with previous findings in the ferret, visually emerges where it is present. Having a conservative threshold for the display would hide most of the pixels (especially considering the small number of trials we need to use) even if they make a coherent map overall (see response to reviewer 3). Importantly, our quantification procedures (decoding and resolution quantification) were controlled for these biases through our randomization procedures. We note here that in the map presented in the paper, we used a 2D median filter (3x3) to filter out isolated voxels (accidentally omitted in the Materials and methods section, now added – we apologize for this omission).

We provide, however, an example image, Author response image 2, with a cluster correction for a family-wise-error rate, with map thresholding based on tuning (as requested by reviewer 3). This correction is based on the comparison between the distribution of spatial cluster sizes in our recordings and those of a random distribution; it is commonly used in fMRI. Our random distribution was generated by mixing the responses across frequencies and trials for each voxel, and we chose the minimal cluster size as the 95th percentile of the random distribution. Only the smallest clusters are excluded, keeping the whole bulk of the auditory cortex intact. We can include this procedure in our manuscript and figures if necessary. The last panel shows a simpler (computationally faster) and coarser method similar to what was presented in the paper, with a median filter (improved to 3D instead of 2D in the paper), as presented in the response to reviewer 3. This last processing step allowed us to coarsely remove artifactual responses that could happen in isolated slices, outside of the auditory cortex.

Applying the same kind of cluster correction to the other thresholding method yielded similar results.

**Author response image 2. respfig2:** 

The exact calculations being made for the resolution quantification need to be written out a little more explicitly and clearly.

We now give more details about the quantification of the spatial resolution. We hope it is now clearer.

For any timeseries analysis involving temporal correlation (whether for external-sensory or electrical stimulation) please describe whether/how temporal autocorrelation was adjusted for in the statistical analyses.

We computed correlation between our responses and the average hemodynamic response function only to decide whether a pixel value should be shown or not (tonotopic map thresholding). Since this was only for visualization purposes, we did not use any adjustment for temporal autocorrelation, in order not to be too conservative on the pixels displayed. We note here that the thresholding used for electrical stimulation experiments was based on simple detection theory (response peak > baseline + 4sem) and not correlation, since less voxels were available and HRF estimation was thus noisier.

Figure 1A caption – presumably the authors mean structural MRI not fMRI?

The reviewer is right. It should be MRI. We corrected this error.

[Editors’ note: the author responses to the re-review follow.]

Although the manuscript is improved, the reviewers had widely differing opinions regarding the extent to which the manuscript has been adequately revised, which we were not able to resolve in the subsequent consultation. In particular, reviewer 3 continued to raise a number of concerns. The key elements of those concerns, along with the minor comments provided by the other reviewers, are summarized in the following. We are hoping that you will be able to address these in a further (and final) revision.

We thank the editor and reviewers for their helpful comments on the description and interpretation of the results. We hope this letter will address the remaining concerns raised by reviewer 3.

Major commentsThe main argument of the authors to justify novelty is that the auditory stimulation (5 tones, 3 s presentation) "pushes the limits of fUS sensitivity". There is no conceptual difference between this paradigm and previous studies that already used fUS to record sensory-evoked activity in rats or other models. The authors argue for shorter stimulation time (3 s), compared to previous studies, as a novel aspect of their paradigm. However, (Urban et al., 2014) already recorded the response to forepaw stimulation as short as 200 µs (see graphical abstract and Figure 4, for example). Yet, this article is not cited. Another claim made by the authors is that they show, for the first time, activations in very small and deep structures. A highly relevant paper (Urban et al., 2015) contains a dedicated section about "Functional imaging in subcortical brain structures", showing, for example, activation in small and deep thalamic nuclei in freely moving rats. However, although this paper was brought to the attention of the authors, they did not add this reference to the manuscript or address these concerns about novelty.

The reviewer here raises two important points that seem to have remained unclear in our last manuscript. We address these issues below and have made the appropriate modifications in the manuscript to clarify them further. Although the reviewer argues that our work is similar to the results presented in Urban et al., (2014,2015), we respectfully disagree. The present manuscript significantly builds upon and beyond the state of the art represented by the Urban et al. papers as we discuss in the following. We highlight several points, either specific to each of the papers or concerning both, that we estimate of high importance to properly estimate the significance of our work.

Urban et al., 2014

Indeed, as the reviewer points out Urban et al., 2014 showed evoked responses to short peripheric electric stimuli. This reference is now included in the manuscript.

The first thing to point out is the difference between ‘normal’ physiological stimuli (tactile, visual, auditory) and the electrical stimulations used in Urban et al., 2014. It is difficult to compare the two kinds of stimuli in terms of speed and efficacy, since in the former, there is a transduction step (with a certain efficiency and temporal scale), in sharp contrast with electrical stimulation which bypasses this stage. The parameters of the electrical stimulation used in Urban et al., 2014 are quite high (200µs, 1mA) and therefore recruit all the group A and probably also group C fibers (Burgess and Perl, 1973; Lebars, 1979). Consequently, the electrical input to the primary cortex is much stronger with electrodes implanted in the forepaw than when trying to stimulate manually the forepaw (stroke or pinching), and the SNR of the vascular response is also higher, enabling very short stimulations. This dichotomy in the stimulation parameters (especially stimulus duration) has also been observed using different techniques (Frostig et al., 1990; Peeters et al., 2001; Lowe et al., 2007; Yu et al., 2012; Berwick et al., 2018).

Secondly, it is not readily apparent from the Urban et al., 2014 paper whether a functional mapping such as ours is even feasible because: (i) they use a much larger number of stimulus presentation (40 trials for each curve, in comparison to 10 in our protocol), (ii) there is a discrepancy in reporting the peak amplitude between Figure 4C (for 1 pulse: 2.7 ± 1.0% peak amplitude) and Figure 4d (~8%), a difference which remains unexplained and casts confusion on these results, (iii) the strong spatial heterogeneity in the evoked responses, with stimulus duration, could compromise the ability of fUS to obtain the functional mapping of whole structures with short, physiological stimuli.

Urban et al., 2015

We had reviewed carefully the Urban et al., (2015) paper, as we mentioned and referred to it in detail in our previous “response to reviewers” letter. We apologize for not having added it to the references of the revised manuscript as it was due to an oversight. It has now been included in the new version. Interestingly, the authors used physiological stimuli in this paper, and showed that reliable% CBV responses could be obtained with short light flashes or manual whisker stimulations. However, it is difficult to estimate from this paper whether short physiological stimuli can be used to reliably activate all brain structures of interest: they do not show any response in deep nuclei to short whisker stimulations (they had to increase the stimulus length to 7s), and no cortical responses to short visual stimuli (Figure 3). Moreover, they used stimuli carefully chosen to trigger the largest responses of the region of interest (e.g., ‘The light stimulus frequency was selected to elicit strong binocular visual responses’), contrary to our protocol, in which stimuli were designed to reveal modulations of CBV response within brain structures.

Consequently, it is hard to conclude from this paper whether our protocol could have allowed us to obtain reliable functional maps.

Issues common to both papers

The reviewer is correct in stating that “functional responses in subcortical nuclei have been observed” previously. We also remarked on this point by citing Gesnik et al., (2017) where they show visual responses in the thalamus and superior colliculus of anesthetized rats. However, these responses were only coarse-grain descriptions of responsiveness to light (flashes) in these structures and not a mapping of their functional organization. Clearly, the same remark can be made about Urban’s 2014 and 2015 papers (e.g., compare Figure 1 in our manuscript with Figure 3 of the Urban et al., 2015). We think that our contributions, demonstrated in the manuscript, towards a finer-grain description of brain organization from very early nuclei to secondary areas are now very clear.

Moreover, beyond the functional maps, we have also quantified the information content of our responses in terms of tuning, thus providing an estimation of fUS signal reliability and sensitivity on small functional scales. No such analysis is found in previous reports.

Finally, we would like to draw attention to the fact that we also developed and effectively demonstrated a new method to speed up the protocol for our single slice analysis (2s tone, and random interval of 4 to 6s – uniformly distributed – between two tone presentations, as described in the Materials and methods section). The method, inspired by classical fMRI analysis techniques, uses a General Linear Model (GLM) to compute impulse responses of individual voxels to each tone frequency, without any predefined hemodynamic response function. This allowed us in turn to present more stimuli (75 repetitions per frequency) in a relatively short time (~45 min), and thus obtain very stable tonotopic maps. While whole structure tonotopic measurements (like many we illustrate in the manuscript) need not rely on this improved protocol, it nevertheless constitutes a notable improvement in task design and analysis over past fUS imaging. Together with the application of MVPA technique, it bridges the gap between fUS and fMRI so as to benefit from the huge methodological advances of the latter.

Reviewer 3 stated that you should emphasize that both ultrasound sequence and data processing used in this study are identical to previous studies, and thus, the imaging resolution remains the same. Indeed, a resolution of 100 µm (the physical resolution of the method) was claimed in previous papers, so it remains unclear why this resolution is presented as "unprecedented". Because the same fUS sequence was used here as in previous studies, the apparent improvements in sensitivity and resolution most likely relate to differences in experimental protocol, including the stimulation strategy used. The revised version of the paper addressed some of these points, but not to the satisfaction of reviewer 3. Please take another look at this and ensure that all relevant key studies are cited.

We agree that the signal acquisition in the fUS system is the same as in former publications, and we did not claim otherwise. We rephrased parts of the introduction in order to clarify our intentions and address this concern. No quantification of the functional resolution has been performed in previous fUS papers. What has been reported before was only the spatial resolution of ultrasound Doppler imaging that is typically expected for ultrasound imaging with a linear array at these frequencies (15MHz): of the order of λ (the wavelength, i.e. 103µm at 15 MHz, with a typical 2 cycles emission for Doppler imaging) for the depth resolution, and of the order of λf/d (with f the depth of imaging and d the physical aperture of the array, i.e. f/d=1 is typically used and achievable for small animal imaging) that is to say 100 µm also for the lateral resolution. This theoretical estimation states that 2 isolated point sources 100 µm apart and reflecting ultrasound signal can be discriminated from each other. This is however a far cry from telling us how finely we can decipher the functional organization underlying the vascular network imaged in that particular ultrasound “Power Doppler” mode. Moreover, the part of the vascular system that is imaged by 2D ultrafast imaging (i.e. slow faster than 1mm/s integrated in voxels of approximately 100 µmx100 µmx300 µm) may itself have a functional resolution larger than 100 µm. What we show here is that by looking at the vascular system with fUS, we can observe different information content at the level of the pixel itself. What we find here is the smallest, lower limit that is reachable with fUS and the characteristics of our protocol (type of stimuli, number of trials, etc.). We managed to do it thanks to the specific design and stimulation strategy (as mentioned by the reviewer/editor), and an in-depth data processing and analysis. In other words, we do not claim to have a *better functional resolution* than previous and recent fUS studies (however we do claim that we obtain a better resolution than other techniques such as fMRI), but we claim to have carefully characterized for the first time this resolution for fUS. We show examples of how useful it can be in capturing the fine-grain organization of brain regions, which has undoubtedly never been done before.

Overall, this technical discussion should not befuddle our findings, which simply show for the first time that a functional mapping of cortical and subcortical brain structures can be performed.

Another concern that has not been well addressed by the authors is their comparison with fMRI. The authors insist on the millisecond acquisition rate of fUS. While the acquisition of a compound ultrasound image (~2 ms) is fast, the relevant information about neural activity is derived from – much slower – images of blood volume (~ 1 s) that are computed from hundreds of ultrasonic images. Thus, the relevant temporal resolution of fUS is on the order of seconds and not few milliseconds. Also, fMRI is not slow compared to fUS; the acquisition of a single slice is usually faster. As an example, (Leaver and Rauschecker, 2016) acquired tonotopic maps using 6 tones of 2 s duration. Each brain slice was acquired in 250 ms and the full volume (28 brain slices) in 7 s. By comparison, a single brain slice is acquired in 1 s with fUS. The advantages of fUS compared to fMRI reside in the higher spatial resolution and ease-of-use, but not in the temporal resolution. Hence, insisting on a superior temporal resolution of the technique is not justified.

As stated in the revised manuscript, the very high compound frame rate of ~5kHz is key in unambiguously discriminating blood motion from tissue motion. Moreover, we note here there is still a lot to explore in the temporal aspect of fUS signal dynamics. We agree with the reviewer that this is not the focus on this paper. We removed the comparison with fMRI in terms of temporal resolution from the Introduction and followed the editors’ advice on insisting more on the higher spatial resolution, ease-of-use and reduced cost and maintenance.

Reviewer 3 continues to have a concern about the way the tonotopic maps were constructed, insisting that this should be based the preferred frequency of pixels modulated by the sound frequency. We believe that – for the reasons that you outlined in the response letter – this is not necessary, but please ensure that the manuscript text fully justifies the thresholding procedures used. We do not think it necessary (or desirable) to use the approach illustrated in the response letter, where a threshold of 10% was used. The reviewer proposed that you should average over more trials or use more stimuli to improve sensitivity to modulation. If you have additional data along these lines, that would clearly help to address this concern, but we believe it is useful to show non-tonopically organized regions too and to focus on the MVPA for quantification of the reliability and sensitivity of the responses (albeit with a little more explanation).

We have now added the necessary justification in the Materials and methods section of the paper: “This thresholding method was used to highlight sound-responsive voxels (disregarding of frequency), and thus display zones that were poorly tonotopic (such as AEG). Note here that this thresholding was used only for visualization purposes. Maps constructed with a threshold based on frequency tuning gave similar qualitative results.”

We agree with the reviewer that having more trials and more stimuli could improve the quality of our results. Increasing the number of trials, for instance, will very likely improve the functional resolution. However, our goal here was to provide a protocol to perform complete tonotopic mappings within a limited amount of time (a few hours only), in order to be able to repeat this procedure daily in behavioral experiments on awake animals. We do not think that having more stimuli or more trials will affect the principal message we want to give to the reader. This study indeed paves the way for more precise developments and does not pretend to reach the highest limits ever reachable with ultrasound imaging. Finally, the fact that within (10 repetitions x 5 frequencies =) 50 trials we can already reach the voxel size as the lower limit for ‘response’ functional resolution seems to indicate that we are already at saturation in that case. Having more trials might only improve the ‘tuning’ functional resolution (300 µm) that is already close to voxel size (100 µm) (Figure 2B, right panels). We present here an example where functional resolution was quantified as a function of the number of trials (average over 5 regions, similar to the ones presented in Figure 2 and Figure 2—figure supplement 3). One can see that both responsiveness resolution and tuning resolution decrease with the number of trials and seem to saturate before 10 trials for responsiveness. We note that part of the noise in the curves stems from the fact that we randomly subsampled the trials.

**Author response image 3. respfig3:** 

We would also like to draw attention to the fact that in one animal (S.), we used 20 trials instead of 10 (as stated in both the Materials and methods section and the supplementary figure captions). This yielded similar results (100/300 µm for the response/tuning functional resolution for the area presented Figure 2—figure supplement 3, lower panel), but the overall responsiveness in this animal was lower (one can observe in Figure 2 a slightly lower decoding accuracy in the AC), suggesting that the number of trials could be adapted to the responsiveness of the regions/animals under study to reach a specific functional resolution. We show here the same figure as above, but on this animal (average over 3 different regions randomly taken from the AC).

**Author response image 4. respfig4:**